# Progresses in Food Packaging, Food Quality, and Safety—Controlled-Release Antioxidant and/or Antimicrobial Packaging

**DOI:** 10.3390/molecules26051263

**Published:** 2021-02-26

**Authors:** Cornelia Vasile, Mihaela Baican

**Affiliations:** 1“P. Poni” Institute of Macromolecular Chemistry, 41 A Grigore Ghica Voda Alley, 70487 Iasi, Romania; 2“Grigore T. Popa” Medicine and Pharmacy University, 16 University Street, 700115 Iaşi, Romania; mihaela.baican@umfiasi.ro

**Keywords:** active food packaging, controlled-release packaging, testing of food quality and safety

## Abstract

Food packaging is designed to protect foods, to provide required information about the food, and to make food handling convenient for distribution to consumers. Packaging has a crucial role in the process of food quality, safety, and shelf-life extension. Possible interactions between food and packaging are important in what is concerning food quality and safety. This review tries to offer a picture of the most important types of active packaging emphasizing the controlled/target release antimicrobial and/or antioxidant packaging including system design, different methods of polymer matrix modification, and processing. The testing methods for the appreciation of the performance of active food packaging, as well as mechanisms and kinetics implied in active compounds release, are summarized. During the last years, many fast advancements in packaging technology appeared, including intelligent or smart packaging (IOSP), (i.e., time–temperature indicators (TTIs), gas indicators, radiofrequency identification (RFID), and others). Legislation is also discussed.

## 1. Introduction

Interrelations between foods and packaging can be detrimental to quality and/or safety. Changes in product flavor due to aroma sorption and the transfer of undesirable flavors from packaging to foods are important mechanisms of deterioration when foods are packaged in polymer-based materials. Selecting packaging materials and developing active, intelligent, packaging is based on a deliberate interaction of the packaging with the food and/or directly improves food quality and safety, in order to maximize product quality, safety, and shelf life, while minimizing undesirable changes. 

Active packaging has the intended effect on the food, which is evidenced by the following processes: advances in delayed oxidation and controlled respiration rate, microbial-growth-inhibiting microorganisms and moisture migration, and the addition of carbon dioxide absorbers/emitters, odor absorbers, ethylene removers, and aroma emitters, while intelligent packaging include time–temperature indicators, ripeness indicators, biosensors, and radiofrequency identification, remove undesirable flavors by sorption, or indicate safety and product shelf life. When a beneficial interaction between the packaging, environment, and food occurs, the bioactive compound is controlled released [1,2], as it is in controlled-release packaging (CRP).

Smart packaging is a term that includes active and intelligent materials. Articles may be placed on the European market if they comply with the restrictions set out in Regulation (EC) 1935/2004, articles, 3, 4, and 15, and the European Regulation (EC) 450/2009. Intelligent packaging provides information on the product or on the product quality and safety. Active-packaging systems are those that include substances deliberately added to the packaging, whose action improves the stability of food quality and safety, or based on the direct contact with the food, provides a condition that improves food stability. Thus, controlled-release packaging is a part of active packaging.

Packaging has a crucial role in the process of food quality, safety, and shelf-life extension. Shelf life of food is defined by Li et al. as “the period of time during which a food retains acceptable characteristics of flavor, color, aroma, texture, nutritional value, and safety, under defined environmental conditions” [3]. Some efficient solutions are known, such as to quickly or instantly add active compounds (such as antioxidants and antimicrobials) as ingredients into the food formulation in a limited concentration or to use active packaging [4,5,6].

During the last years, many advancements in packaging technology appeared, including intelligent or smart packaging (IOSP) (i.e., time–temperature indicators (TTIs), gas indicators, radiofrequency identification (RFID), and others), and active packaging (AP; such as oxygen scavengers, moisture absorbers, and antimicrobials) [7] (see paragraph 7). These innovations are responsible for the improvement in food quality, safety, and shelf life.

## 2. Types of Active Packaging

There are several designs for active packaging (AP) inside a food package (Figure 1), such as the following: (A) using the active sachets (oxygen and moisture absorbers, and ethanol vapor generators), (B) coating an active compound (heat-sensitive active agents or those incompatible and immiscible with the polymer matrix) onto the polymer, (C) immobilizing the active compound on the polymer surface (the presence of functional interacting groups on both the active agent and the polymer is necessary). The strong bonding of active compounds onto polymers allows slow release into the food, and (D) direct incorporation into the polymer matrix, which ensures high resistance to processing conditions of the polymer, no adverse effect on the polymer properties, and slow release to food. Use of bioactive polymers, such as chitosan, exhibits inherently antimicrobial activity in composites or coating (E) [8,9].

According to the mechanism of action, the AP systems are divided into three types, including the following: “releasing systems” (e.g., antimicrobials, antioxidants, CO_2_ or ethanol emitters, or enzymes), “absorbing systems” (e.g., oxygen, CO_2_ or ethylene scavengers, moisture, or aroma absorbers), and “nonmigrating system”, where full contact of food and AP is required (e.g., enzymes acting on the surface of food, their migration being not necessary, and antimicrobial nanoparticles (NPs), such as nanosilver, titanium dioxide (TiO_2_), and zinc oxide (ZnO)). NPs can be used as antimicrobial agents, but their migration to the food is not allowed by laws [11]. The active-releasing systems can be also divided into: “leaching systems”—the releasing of a substance achieved by direct contact between food and packaging material (e.g., those by bacteriocins, antibiotics, antioxidants, polyphenols, or organic acids), and “volatile systems”—releasing occurring through gas-phase diffusion from the packaging layer to the food surface. This type of releasing is applicable only for volatile active agents (essential oils (EOs) or low-molecular-weight bioactive compounds, such as menthol, carvacrol, and linalool (“a naturally occurring terpene alcohol chemical found in many flowers and spice plants with many commercial applications, the majority of which are based on its pleasant scent”, cinnamaldehyde).

Active-releasing antimicrobial packaging applications are directly related to food microbial safety, as well as to shelf-life extension, by preventing the growth of spoilage and/or pathogenic microorganisms (because the growth of spoilage microorganisms can not only reduce the food’s shelf life, but it can also endanger public health, particularly in the case of pathogenic microorganisms). However, both antimicrobial resistance [12] and pro-oxidation in lipid foods can be developed [13]. The newest solution is controlled-release packaging (CRP) [14].

## 3. Controlled-Release Packaging (CRP)

The fundamental concept of CRP is to use a package with a modified concentration of an active compound surrounding the food product inside the package, in order to retard the deterioration of the food and extend its shelf life. This concentration may change with time, following a “concentration profile”. Such packaging can release active compounds at different controlled rates, suitable for enhancing both quality and safety of foods during extended storage. It aims not only to prolong the duration of active compound delivery but also to promote the predictability and reproducibility of release rates. The package in CRP systems serves as a carrier to incorporate and retain the active compound (AC) into its composition and to release it at an appropriate time. Retaining the AC inside the package takes place in a limited space; therefore, the tortuous paths for the AC are trapped/retained inside composition or by a barrier layer or coating. Loading, retardation, and releasing of the active compound from the package composition can be influenced by external factors, such as moisture, heat, or removal of the barrier layer, or by the intermolecular forces between the AC and the package. From a commercial point of view, a changing concentration with time has practical value in some situations, such as when it is desirable to have a higher concentration of antimicrobials or antioxidants in the beginning to retard microbial and oxidative deterioration in order to extend shelf life of the product during distribution and storage, while having a lower concentration when the product reaches the consumer. CRP is a new generation of functional packaging, acting as a delivery system for the AC release (antimicrobials, antioxidants, flavors, aromas, and plant growth hormones, such as 1-methylcyclopropene, to affect fruit ripening; initiation or delay) in a controlled manner to improve safety. Retaining quality for a wide range of food products during storage was defined even in antimicrobial packaging and antioxidant packaging, but the terms are different [15,16]. The AC is released in a timely manner to enhance safety, retain quality, and extend the shelf life of food [15]. In the Federal Register of July 17, 1995, (60 FR 36582), the US Food and Drug Administration (FDA) established a “threshold of regulation” release process. This process was established for determining when the extent of migration of some additives to food is so trivial that safety concerns would be negligible. The process exempts materials in food-contact articles whose use results in dietary concentrations situated at or below 0.5 ppb (μg/kg) from the food-additive-listing regulation requirement. Carcinogens or substances that may be carcinogens are excluded from this regulation. Careful consideration must be given to those factors affecting such interactions when selecting packaging materials in order to maximize product quality, safety, and shelf life, while minimizing undesirable changes. The key safety objective for traditional materials in contact with foods is to be as inert as possible, i.e., there should be a minimum of interaction between food and packaging. Meanwhile, as a response to consumer demands or industrial production, new food-packaging technologies developed trends toward mildly preserved, fresh, tasty, and convenient food products with a prolonged shelf life and controlled quality. Product considerations include sensitivity to flavor and related deteriorations, color changes, vitamin loss, microbial activity, and amount of available flavor. Storage considerations include the temperature, time, and processing method. Polymer considerations include the type of polymer and processing method, volume or mass of polymer-to-product ratio, and whether the interaction is Fickian or non-Fickian [17].

Gas chromatographic methods for the detection of volatiles and new techniques involving supercritical gases for the analysis of minimally volatile constituents of both specific and overall migration are proposed and used. The organoleptic properties and the average migration potential in different groups of materials should be investigated. Better correlations between rates of migration into food simulants and into real foodstuffs should be found by theoretical predictions of migration based on empirical data for partition and diffusion coefficients in order to make the evaluation of plastics [18,19].

Quality loss and microbial safety are major concerns of the food industry. It is known that packaging plastics contain substances of low or medium molecular weight. Residual monomers and additives such as antioxidants for plastics or lubricants may migrate into the packed good after filling and may form undesirable levels of concentrations in the product until consumption. One main purpose of migration testing is to ensure product safety and protect the consumer by controlling the levels of such constituents from the packaging in the product. Therefore, the manufacturers of packaging materials must assure that their quality is safe for food and drug applications [20].

The uniqueness of CRP is that it focuses on the kinetics and mechanism of controlled release. A CRP system may contain two antimicrobials for different target microorganisms [21], or it may contain both antimicrobial and antioxidant to inhibit microbial growth and lipid oxidation [22], which can have many possible “release rate profiles,” defined as plots of rate of release of AC versus time. AC is obviously the first design factor, and its selection depends on efficacy, regulatory compliance, and cost. For most CRP systems, the release rates change with time. However, when an AC is incorporated into a film, its release is both constant or often governed by diffusion of the AC in the film, following the diffusion-controlled release-rate profile, characterized by fast release initially, with a progressively slower release as time passes. The recommended time to start release is immediately after filling the food and sealing the package. At a slow rate, an insufficient amount of AC is released to retard food deterioration, while at a fast rate, an excessive amount of AC is released.

The application of CRP has been widely and effectively used in pharmaceuticals and drug industries, but its application in food industry is relatively new. An example of a CRP application in food which already exists in the market is the use of butylated hydroxytoluene (BHT) as an antioxidant to extend the shelf life of breakfast cereal [23], but its use is still debated [24].

### 3.1. Active Compounds

Active properties can be conferred by the incorporation into the packaging materials of substances with inherent antioxidant and antimicrobial properties [25]. Common active compounds used in CRP include mainly antimicrobials for food safety and antioxidants for food quality, oxygen, or ethanol scavengers, and CO_2_ emitters.

***Antimicrobials.*** An antimicrobial is an agent that kills microorganisms or stops their growth. Antimicrobial packaging is multifunctional by reducing harmful microbial activity in food, helps to increase food safety, reduces food wastage, and improves food shelf life. Biobased antimicrobial agents in packaging provide extra safety for health [26]. Those for food preservation act to prevent growth of spoilage and pathogenic microorganisms. Antimicrobial agents are incorporated into polymer film/packaging to suppress the activities of targeted microorganisms, as against *Listeria monocytogenes, Mycobacterium smegmatis (MTCC 943), Pseudomonas aeroginosa (MTCC 4676), Escherichia coli* O157, *Salmonella, Staphylococcus aureus, Bacillus cereus, Campylobacter, Clostridium perfringens, Aspergillus niger*, *Saccharomyces cerevisiae*, etc. Between the most-used antimicrobials, one can mention the following: main natural compounds, as essential oils derived from plants (e.g., basil, thyme, oregano, cinnamon, clove, and rosemary), enzymes obtained from animal sources (e.g., lysozyme and lactoferrin), bacteriocins from microbial sources (nisin and natamycin), organic acids (e.g., sorbic, propionic, and citric acid), naturally occurring polymers (chitosan and its derivatives) [27], sodium benzoate, etc., all approved to be used in contact with food [28].

Interest in “antimicrobial activity” of chitosan is hugely evidenced in 2014 by over 1140 articles, with 740 of these published after 2010. Derivatization of chitosan is realized by acylation, carboxylation, alkylation, and quaternization in order to improve the water solubility, pH sensitivity, and the targeting in the antibacterial, sustained slow release, targeting, and delivery system fields. Chitosan derivatives present excellent antimicrobial activity due to permanent positive charge on nitrogen atoms side-bonded to the polymer backbone [29,30,31].

Other interesting polysaccharides used for antimicrobial packaging are alginates and carrageenan [32]. Nano-formulations of silver nanoparticles with cellulose, chitosan, and alginic-acid biopolymers for antibacterial applications have been prepared [33]. Incorporation of silver NPs as nanocomposite (NC) forms improved antibacterial activities of these polysaccharides. Antimicrobial packaging with lactic-acid bacteria incorporated in alginate film matrix was found to control the growth of food-borne pathogens in ready-to-eat food [34].

They are affected by a variety of intrinsic factors, such as pH and presence of oxygen, or by extrinsic factors associated with storage conditions, including temperature, time, and relative humidity. They can also include flavors, aromas, enzymes, probiotic bacteria, nutraceuticals, and plant growth hormones (such as 1-methyl cyclopropene to delay fruit ripening). Some active compounds are multifunctional, such as some essential oils (EOs) with antimicrobial, antioxidant, and antifungal activity [35,36,37]. Their release from packaging is determined by volatility, molecular shape, size, polarity, and weight [38], and the presence of the other bioactive compounds [39]. Antimicrobial agents from organic sources (such as plant phenolics, carvacrol, thymol, bacteriocins) and monoterpene hydrocarbons (p-cymene and γ-terpinene) compounds which are present in oregano essential oils, citric acid (in green tea), (clove, oregano, and thyme), enzymes (lysozyme, lactoperoxidase, and glucose oxidase,), polymers (chitosan and derivatives), organic acid (acetic acid, lactic acid, and benzoic acid), bacteriocins (nisin), and metal ions (zinc oxide zinc, silver nanoparticles, copper, palladium, and titanium more stable at higher temperatures) were incorporated into the protein-based films [40]. Nowadays, protein-based film technology has emerged as one of the most extensively studied in the food-packaging sector, as it exhibits good mechanical, optical, and oxygen barrier properties. In addition, protein-based film also showed good compatibility to polar surfaces, while having effective control on the release of additives and bioactive compounds in the food-packaging system. Antimicrobial food packaging gained great interest due to high inhibition of microbial activity that helps in prolonging the shelf life of packaged food and enhancing the food’s safety, while improving the functionality of the films. They are also biodegradable. However, biocomposite protein-based packaging films obtained by incorporation of antimicrobial agents might also require chemical, toxicological, and further tests to secure more safe and approved products according to the standard food safety regulations while being able to deliver good means in protecting the safety and quality of packaged food.

***Antioxidants***. Polyphenols, flavonoids, (e.g., quercitrin), vitamins, poly-unsaturated fatty acids, curcumin, astaxanthin, catechins, selenium nanoparticles (SeNP), resveratrol, etc., show antioxidant properties and are used in food products. They act both as chain-breaking antioxidants or as hydroperoxide-deactivating antioxidants. Reactive oxygen species (ROS) (such as: O_2_•, HO•, HO_2_•, RO•, ROO•), or reactive oxygen-containing compounds (such as: H_2_O_2_, O_3_, ^1^O_2_) show various reactivity and oxidizing ability, participating in diverse chemical reactions (oxidative stress) and producing decomposition of biologically active compounds or biomolecules. Antioxidants protect other molecules from the damaging effects of such ROS and are used as AC in formulations preventing oxidative stress [41]. They are appropriately encapsulated when there are favorable interactions between the functional groups of the encapsulated compound and the encapsulating material/nanomaterial. The main nanoencapsulation techniques applied to antioxidants and antimicrobials are described: association colloid-based nanoincorporation, lipid-based nanoencapsulation techniques, encapsulation techniques based on biologically derived polymeric nanocarriers, encapsulation techniques based on nonbiological polymeric nanocarriers, cyclodextrin incorporation, electrospraying and electrospinning, carbon nanotubes, and nanocomposite encapsulation [42]. Several nanoencapsulation methods can be followed by freeze-drying or spray-drying. An improvement to aqueous solubility, antioxidant, and other health-promoting properties, in vitro gastrointestinal (GI) release profile, and protection against process and environmentally harsh conditions (e.g., light, oxygen, high temperatures, and humidity) of hydrophobic food bioactive compounds could be achieved by nanoencapsulation, using different nanoencapsulations, including inclusion complexes of CDs, amylose, yeast cells, nanogels, natural extracts (NEs), nanofibers, nanosponges, nanoliposomes, and NPs made with lipids [43], see also below. As an example, there could be the aforementioned astaxanthin-loaded nanostructured lipid carriers with the Z-average size of 94 nm containing α-tocopherol and EDTA as antioxidants, which were stabilized using Tween 80 and lecithin and mixed with nonpasteurized CO_2_-free beer at the volume ratio of 3:97; these showed improved stability at a low storage temperature of 6 °C [44]. Ethyl cellulose microparticles with encapsulated hydroxytyrosol, a constituent of olive oil showing antioxidant properties, demonstrated the effectiveness of their gastroresistance and the antioxidant capacity preservation of >50%, indicating possible applications of this formulation in foods, drugs, and nutraceuticals [45]. Nanoliposomes incorporating olive-leaf extract with high levels of phenolic compounds and oleuropein, showing antioxidant and antimicrobial activities, with average particle size 25–158 nm, negative charge, were supplemented to yogurt, which improved its antioxidant activity, and no significant changes in color and sensorial attributes were observed, suggesting that olive-leaf phenolics can be entrapped in nanoliposomes and could increase the nutritional value of products like yogurt [46]. The aqueous solubility of resveratrol (RES) from α-lactalbumin (α-Lalb)-RES nanocomplexes was 32-fold higher than that of free RES, and the nanocomplexes considerably improved the antioxidant chemical stability under storage, at pH 8.0 and high temperature, and showed very good in vitro antioxidant activity compared to free RES, suggesting that α-Lalb as a nanoscale carrier could effectively deliver lipophilic nutraceuticals in the functional food, biomedical, and pharmaceutical products [47]. Digested kenaf (*Hibiscus cannabinus* L.) seed O/W (nanoemulsion) natural extracts (NEs), stabilized by a SCas, Tween 20, and β-CD complex, show a good bioaccessibility of antioxidants (tocopherols and total phenolic contents) and a lower phytosterol degradation rate compared to digested bulk oil, which indicates the possibility of their future application in food and nutraceutical industries [48]. Many spices and herbs are recognized as sources of bioactive compounds which are able to stabilize free radicals and prevent oxidation processes and/or act as bacteriostatic or bactericidal agents [49,50], such as essential oils (EOs) (as oregano essential oil, natural extracts (NEs) (rosemary extract, green tea extract (GTE), sage extract) and/or inorganic and metal nanoparticles. Essential oils and natural extracts such as *Salvia officinalis* are generally recognized as safe according to the U.S. Food and Drug Administration (FDA) [51]. EOs are volatile compounds obtained from aromatic plants that produced them naturally as secondary metabolites. They also exhibit important antimicrobial activity in the vapor phase, being applied to bakery products, because they are able to delay the microbiological spoilage [52]. EOs and NEs are mainly composed of terpenoids, phenolic, and aromatic compounds, and their composition can widely vary depending on the edaphoclimatic characteristics of the plant, the part of the plant (i.e., flower, seed, leaves, fruits, stems, and others), and the extraction procedure [53,54]. There is great interest in the use of these natural products, because they are classified as generally recognized as safe (GRAS) food additives by the FDA [55]. Essential-oils extracts are multifunctional, because they can be used as both antioxidants, antifungals, and antimicrobials. They are increasingly studied as solutions to replace synthetic ones [56,57,58,59]. Active films were developed by means of the electrospinning technique and subsequent annealing treatment based on poly (ε-caprolactone) (PCL) containing a solid dispersion of multifunctional sage extract (SE, 5–20%). The water vapor and aroma permeability of the obtained films increased by adding SE to the polymer; by SE incorporation into PCL matrix. A strong 2,2-diphenyl-1-picrylhydrazyl (DPPH·) free radical scavenging ability and a strong activity against foodborne pathogens such as *Staphylococcus aureus and Escherichia coli* [25] were also obtained. Both volatile and nonvolatile antimicrobials may be used in CRP, where there is direct food/package contact (e.g., in a vacuum-packed fresh-meat pouch), because there is either a migration of microbes to the antimicrobials or vice versa [60,61]. Only volatile antimicrobials may be used where there is no direct food/package contact (e.g., in a fresh-produce bag where there is air space between the produce and the bag) [62,63]. For antioxidants, the situation is different from antimicrobials: nonvolatile antioxidants encapsulated in the package can be effective for oxidation inhibition where there is no direct food/package contact. This is because free radicals generated from oxidation and oxygen in the package can travel through space to reach the nonvolatile antioxidants in the package [64,65]. CRP containing volatile antioxidants as sesamol is also effective; for example, for lipid oxidation of oat cereals [66]. The mobility of volatile compounds explains their higher efficiency when compared to nonvolatile compounds for retarding microbial growth and oxidation. However, this mobility can also cause undesirable loss of more volatile compounds such as butylated hydroxytoluene (BHT) during processing (such as in a film-extrusion process where high temperature and shear are involved) and during storage, due to premature release of the volatile compound than of the nonvolatile tocopherol [67]. By protecting the volatile compounds by encapsulation and barrier layer, the losses are diminished. Another requirement is the compatibility of the active compound with the packaging material. This affects both loading and release of the active compound from the packaging material with antioxidants [39,68]. Release can be modified (slow or fast) by encapsulation. The encapsulated tocopherol into γ-cyclodextrin and transformation into nanofibers with polylactic acid (PLA) by electrospinning had a faster release rate for the system with γ-cyclodextrin encapsulation than the system without encapsulation, because γ-cyclodextrin improved the tocopherol solubility in the aqueous phase of the food simulant. This was demonstrated for other system, too, as lecithin nanoencapsulated nisin [69]. Hydroxypropyl methyl cellulose (HPMC) film incorporating PLA/green-tea extract [70] and polyethylene (PE) film incorporating green-tea extract [71] are also used for food-packaging applications.

### 3.2. Package Composition and Structure

Both synthetic polymers and biobased polymers and also their blends have been studied as packaging films. In the last decade, the research was focused to biobased polmyers, because they are more environmentally friendly than synthetic polymers. Some polyhydroxyalkanoates (PHAs), such as poly(3-hydroxybutyrate) (PHB), poly(3-hydroxybutyrate-co-3-hydroxyvalerate) (PHBV), poly(3-hydroxybutyrate-co-4-(P(3HB-co-4HB)), and poly(3-hydroxybutyrate-co-3-hydroxyhexanoate) (PHBH) are currently frequently studied to develop bioplastic packaging articles, such as injection-molded pieces, compression-molded sheets, and films [72,73]. The structure and morphology of biobased polymers can be modified by crosslinking, processing, and other methods to obtain various release rates for active compounds (e.g., the change in cellulose film morphology from dense to porous or the release rates of L-ascorbic acid, tyrosine, and lysozyme) [74,75]. The composition of stereochemical isomers of PLA [14], as well as the processing methods including drying, annealing, solution casting, and extrusion, and the crystallinity of the resulted film can be controlled. A wide range of tocopherol release rate profiles can be obtained, this one controlling the degree of crosslinking in low methyl pectin film with calcium ion as a crosslinker to change release rate and the released amount of nisin [76] in different combinations, such as a blend of chitosan and cellulose incorporating sodium benzoate or potassium sorbate [77], and blend of chitosan and konjac glucomannan incorporating nisin [3,21]. PLA/thymol; PLA/BHA, BHT, PG and TBHQ/methylcellulose edible film, Ferulago angulate/essential oil nanocapsules; zein film/lauryl arginate (LAE); cassava starch film/bixin; pectin/carboxymethyl cellulose films/potassium sorbate; whey protein film/essential oil; starch film/BHT; starch film / green tea extract, soy protein film/incorporating /nisin, and PLA-PHB film/catechin [15], as well as blends of LDPE/ chitosan and PLA/essential oils, have been also studied [21]. Polymer blends were used to improve physical properties of packaging films.

Multilayer film structure is studied very often as CRP, because the layers may have various functions and desirable properties, such as high barrier, mechanical strength, or heat sealability that no single material possesses. The multilayer active films are composed of three layers: the barrier layer (with high barrier properties, preventing the loss of active substances to the environment), matrix layer (containing active substance and showing very fast diffusion), and control layer (with lower swelling ability than matrix layer, controlling the release of the active agent to the food) [78]. The thickness, chemical composition and diffusivity of the control layer must be varied with each type of food. Multilayer films are produced by co-extrusion method, layer-by-layer (LbL) deposition method [79], lamination, coating, co-injection with stretch blow molding, etc.

Commonly, ACs are contained inside a surface layer in direct contact with food. LDPE/4% tocopherol, coextruded with high density polyethylene (HDPE), and ethylene vinyl alcohol (EVOH) have improved water vapor and oxygen barrier properties, being used as a milk-powder package [80]. Coating technology is applied to obtain an active surface layer which can be disintegrated over time, when it is physically deposited, but this behavior is not observed when it is covalently bonded [81,82,83,84,85,86,87]. If the active compounds are placed into the core layer, they are protected from oxidation.

Some examples of CRP with multilayer structure are summarized in Table 1.

Barrier properties (water vapor transmission rate, gas transmission rate, light, flavor, and aroma) of the multilayer packaging films are superior when compared to those of the monolayer packaging films. This behavior with has direct implications on the shelf life of packaged foods. During the process of multilayer packaging testing, determination of the physical, chemical, and mechanical properties of the polymers present in the respective system is required, such as [104] seal strength evidenced by the peel test; gas and water vapor permeation evidenced by oxygen transmission rate (OTR); stiffness evidenced by the bend under own weight; tear strength evidenced by the cantilever test, trouser-tear method, or Elmendorf tear-resistance test; scratch and abrasion resistance evidenced by scratch tests; and adhesion evidenced by the peel test, etc.

For instance, EVOH is a copolymer widely used in food packaging for its excellent gas-barrier property. However, its use can be limited because of the presence of moisture-sensitive hydroxyl groups. This disadvantage can be overcome by forming a multilayer structure using hydrophobic materials such as PP or LDPE [105].

PVA/vermiculite nanocomposite-coated multilayer packaging film was tested for storing high-moisture foods using three food stimulants (deionized water, 3% acetic acid, and 50% ethanol) [106]. It was observed that oxygen permeability significantly increased for a relative humidity of about 60%, with no deterioration in oxygen-barrier properties with deionized water and acetic acid but structural changes with ethanol.

### 3.3. CRP System Design

The main CRP objectives are to manipulate polymer morphology to achieve the required release kinetics (fast or slow release) to match the food type, while antimicrobial and antioxidant packaging are nonreleasing, being like those grafted onto packaging materials, as well as nonreleasing systems involving oxygen absorbers and free radical scavengers. The chemical modification of polymers, the preparation of multilayer films, and the use of crosslinking agents are some methods tried in the last decades. Other approaches use molecular complexes and irradiation treatments. Micro- or nano-encapsulation of active compounds and using nanostructured materials in the CRP film matrix are the newest techniques used for the preparation of CRP systems [10].

Active compounds type is obviously the first design factor, and its selection depends on efficacy, regulatory compliance, and cost. The second design factor is package composition and structure that form a matrix or carrier, such as a packaging film or a coating, to encapsulate the active compound. The third design factor is the processing methods, such as cast film extrusion and blown film extrusion, which serve the purpose of incorporating the active compound into the packaging polymer and creating an appropriate morphology or structure to allow the release of the active compound. It was applied for HDPE, PP, and their mixtures. Modifying the processing variables and applying new techniques methods, accomplished by smart blender and polymer blends, led to changes in diffusivity; hence, different release profiles were achieved. Biodegradable polymers such as pectin, gelatin, chitosan, polylactic acid, and others are suitable components.

Poly(L-lactic acid) (P-LLA) and poly(D-lactic acid) (P-DLA) affect polymer final properties and provide a wide range of release profiles. The delivery occurs in a timely manner to increase shelf life and maintain safety and quality of the food [14]. PLA containing different ratios of stereochemical isomers was impregnated with 3000 ppm tocopherol using solvent casting and commercial scale extrusion or casting methods to produce the films. The solution casting films were first dried at room temperature for 24 h and further dried at 40 °C. Results showed different tocopherol release profiles and diffusivities (2.42 × 10^−19^ to 8.68 × 10^−16^ m^2^/s) with three orders of magnitude, while the annealing process increased film crystallinity, which led to lower diffusivity and slower tocopherol release and decreased water vapor and oxygen transmission rates of the films. Composition of the packaging materials is also an important parameter.

### 3.4. Micro- and Nano-Materials

Microcapsules are particles with a diameter between 3 and 800 μm, while nanocapsules are in the range of 10 to 1000 nm (1 μm) [103,107]. Both microcapsules and nanocarriers can protect a bioactive compound from unfavorable environmental conditions, e.g., oxidation, heat, light, pH, and enzyme degradation. Nanotechnology by new nanoformulations provides new or modified properties conferred to many current materials.

Nanomaterials can be incorporated in food-packaging design by melt compounding, solvent casting, lamination, or electrohydrodynamic processing (EHDP), in the form of nanocomposites using as carriers’ biopolymer matrices. They offer passive, active, and even bioactive properties (barrier, antimicrobial, antioxidant, and oxygen scavenging) and also have roles and the controlled release of functional ingredients exerted either by the intended or non-intended migration of the nanomaterials or by the active substances they may carry [108]. In general, the CRP systems utilized in the food industry can be classified into two classes based on the main applied materials [109]: (1) lipid-based nano-carriers formulated by lipids (fats and oils) and (2) biopolymer-based delivery systems, such as polysaccharides (starch, pectin, chitosan, cellulose derivatives, gum arabic, guar gum, alginates, etc.), proteins (whey protein concentrate and isolate (casein, gelatin, soy protein isolate, etc.), or their mixtures.

Lipid-based nanocarriers can entrap materials with different solubility, can be produced in industrial scale, are capable of targeted and triggered release control, but show some adverse effects on color, taste, and aroma of foods and beverages. Biopolymeric systems are suitable carriers for high-temperature processes, show higher loading capacity, stronger protection of encapsulated compounds, and a lower release rate. The lower interaction of active compounds with polymer matrix and their nano-encapsulation to protect their chemical structure are reasons for an increased activity after migration and decreased adverse effects of additives, as color and appearance of the active films.

Some examples of nanocompounds used in CRP are bacterial cellulose nanocrystals [110], cellulose nanofillers as cellulose nanofibers and other nanocelluloses used to prepare cellulosic biodegradable composites [111], and other natural polymers such as nanoparticles (NPs). As inorganic nanoparticles, silver nanoparticles (AgNPs) [112], copper oxide nanoparticles (CuONPs) [113], zinc oxide nanoparticles (ZnONPs) [114], mesoporous silica nanoparticles [115], incorporated in electrospun poly hydroxy alkanoates (PHA), or its copolymers materials and other polymers are used. Selenium nanoparticles (SeNPs) with a size between 50 and 70 nm, incorporated in laminates as colorless multilayer (e.g., polyethylene terephthalate (PET, 12 μm), 3 g/m^2^ adhesive, and a second 20 × 30 cm of 60 μm thickness film of low-density polyethylene (LDPE)), can prevent the oxidation and extend the shelf life of hazelnuts, walnuts, and potato chips [116]. These multilayers were also tested for ready-to-eat vegetable mixtures seasoned with butter, consisting of sweet corn, chopped broccoli, and chopped carrot and fresh raw chicken breast. Metalloid nanoparticles incorporated into a PET/adhesive/LDPE multilayer perform a strong free radical scavenger activity. In such active-packaging materials, the nanomaterials are not in direct contact with the food, but they protect the food, reducing the formation of lipid off-flavors. Poly(3-hydroxybutyrate) (PHB) containing palladium nanoparticles (PdNPs) films, pretreated with surfactants, and permitted for food-contact applications, had high oxygen scavenging performance, water barrier, and oxygen [117]. Scavenging multilayer films were developed by coating paper with electrospun PdNP-containing PHB and poly(ε-caprolactone) (PCL) fiber mats, as paper multilayer exhibited higher oxygen scavenging capacity [118]. Mineral nanoparticles containing 50 wt% of eugenols were loaded into PHBV by electrospinning [115], followed by annealing. There were thus produced continuous films, which inhibited bacterial growth. Materials such as interlayers or coatings for active food packaging applications showed enhanced barrier properties, due to the presence of the eugenol-containing nanofillers, at optimal contents around 15 wt %. Their antimicrobial activity against *S. aureus* and *E. coli*, in both open and closed systems (which represent the real conditions in packaging applications) of such electrospun biopolymer films, showed antibacterial activity after 15 days, being higher (as expected). This behavior was ascribed to the accumulation of eugenol in the system’s headspace.

Encapsulated oregano essential oil (OEO), rosemary extract (RE), and green tea extract (GTE) were incorporated in ultrathin fibers of poly(3-hydroxybutyrate-co-3-hydroxyvalerate) (PHBV) derived from fruit waste, using solution electrospinning, and the resultant electrospun mats were annealed to produce continuous films [73]. The obtained films presented the highest antimicrobial and antioxidant activity against the food-borne bacteria *Staphylococcus aureus (S. aureus)* and *Escherichia coli* (*E. coli*). This activity can be explained by their composition. OEO mainly contains isomer phenols, carvacrol and thymol, RE phenolic, and the volatile constituents carnosol, carnosic acid, and rosmarinic acid, while GTE is mainly composed of gallic acid, theobromine, chlorogenic acid, and caffeic acid. The electrospun OEO-containing PHBV films presented the highest antimicrobial activity against two strains of food-borne bacteria, as well as the most significant antioxidant performance, which was ascribed to the films high content in carvacrol and thymol. Such electrospun/annealed PHBV films can be applied as active layers to prolong the shelf life of the foods in biopackaging applications. In the frame of a circular bioeconomy, such materials prolong the shelf life of foods and delay the proliferation of microorganisms and enzymatic oxidation.

Nanoformulations are used in the food industry for various types of nutritional supplements, prepared to improve bioavailability, and obtaining functional foods, edible products (e.g., food, food constituents, or supplements), to protect active ingredients against degradation or to reduce side effects [119]. Nutritional supplements are compounds of natural origin, such as curcumin (CUR) occurring in turmeric, Ω-3-fatty acid in fish oil, vitamins from fruits, probiotics, etc., as active ingredients which, when encapsulated in an appropriate nanocarriers (for encapsulating nutraceuticals and fortification of food products—nanostructured lipid carriers), are released after consumption of the food in the target organ and utilized according to its nutritional properties [120]. Nanocarriers can be nanogels, nanosponges, core-shell nanoparticles (NP), nanofibers, cyclodextrin complexes, mesoporosous silica NP, core-shell NP, layered double hydroxides, nanoemulsions, micelles, etc. The intelligent dietary supplements for bioactive compounds in foods are designed to improve their low solubility, poor stability, and low permeability in the GI tract and also to increase their oral bioavailability. Food dietary supplements are used to create food for infants, follow-on nourishment, and nutrition of small children, low-energy foods designed to reduce body weight, gluten-free foods, foods for people with disorders of carbohydrate metabolism (diabetics), low lactose or lactose-free foods, foods with low protein content, foods intended for athletes and for persons with increased physical performance, etc. Foods for special medical purposes (FSMPs) are advised to be used only under medical supervision and have to be provided with labels with information about their intended use. A functional food or a functional ingredient is any food or food component providing health benefits beyond basic nutrition, and natural bioactive compounds in low concentration. Functional ingredients showing beneficial effects for health have become increasingly popular in the diet [121]. Therefore, functional foods are similar to traditional conventional foods but have more advantageous properties in relation to healthy physical condition. The Pluronic^®^ modified curcumin liposomes showed a slower release rate and lower cumulative release percentage for curcumin, enhanced pH stability and thermal stability, and a significantly improved absorption in simulated GI tract in vitro, suggesting that both types of liposomes could be used as carriers [122]. Nanocapsules, based on lipid formulations having larger surface areas than microsized carriers, can more effectively enhance solubility, bioavailability, and controlled release of nanoencapsulated phenolic compounds and could be successfully applied in functional foods [123]. Nanostructured lipid carriers, such as micro- and nano-biobased delivery systems for food products, such as vitamin A palmitate-loaded [124], solid lipid microparticles (MPs) loaded with 0.1% of vitamin D3 [125], etc., have been prepared. Polysaccharide-based carriers show various enzymatic susceptibilities and specific degradation in intestines. Targeting release has been obtained, such as low-molecular-weight octenyl succinic anhydride modified starches which are suitable to form stable vitamin E nanocapsules for potential application in beverages [126] and cellulosic nanomaterials for food and nutraceutical [127]. The following nanomaterials for food products have been developed: cellulose nanocrystals and lecithin into alginate microbeads [128], oligo-hyaluronic acid-CUR polymer co-loaded with both curcumin and resveratrol, applicable in juice, yoghourt and nutritional supplements [129], chitosan (CS)/tripolyphosphate-nanoliposomes core-shell nanocomplexes, as vitamin E carriers [130], CS hydrochloride/carboxymethyl CS nanocomplexes loaded with anthocyanins [131], food-grade alginate/CS nanolaminates with incorporated folic acid [132], multifunctional stimuli-responsive NPs, based on chondroitin sulfate/bovine serum albumin [133], protein-polysaccharide (propylene glycol alginate)-surfactant ternary complex particles [134].

Functionalization of cellulose materials through surface modification is very important for biomaterials intended to be used in food preservation, because surface functional properties of cellulose-based composites is a desirable criterion to enhance its barrier capability against moisture, water vapor, and gases exchanges between the food and conditional atmosphere. This enhancement in the barrier properties could protect the food via reducing the degrading rates from physical, chemical, and microbiological changes [135].

Razavi et al. [136] demonstrated that nanoparticles of zinc oxide (ZnO) play a major role in the antibacterial properties of cellulose-based film. A chitosan solution containing zinc oxide nanoparticles was used for packaging cheese and chicken breast meat, proving an enhancement in the antimicrobial properties against *L. monocytogenes*.

Cellulose acetate nanocomposite film with silver nanoparticle was very efficient in suppressing the growth of *B. cereus*, *E. coli*, *K. pneumoniae*, *S. typhi*, and *S. aureus* [137]. Silver nanoparticles were also incorporated into banana-peel powder-filled cellulose matrix [138], the obtained hybrid composite showing good inhibitory activity on gram-negative (*P. aeruginosa* and *E. coli*) bacteria when compared to gram-positive (*B. licheniformis* and *S. aureus*) bacteria.

*Protein-based carriers.* Protein-based nanoencapsulation extend their functionality in nanocarrier systems to achieve an improvement in encapsulation, retention, protection, and release of bioactive agents [139,140], which have activity dependent on pH; for example, one can mention folic acid binding to β-lactoglobulin (β-Lglb), and type A gelatin carriers [141], 7S and 11S globulins (Glbs) obtained from soy flour complexed folic acid [142]. Encapsulation of proteins into microvesicles of bacteria, generally considered as safe, could be also used in applications of foods and nutraceuticals. In vitamin D–potato protein co-assemblies, the nanocomplexation provided pronounced protection and reduced vitamin D losses during pasteurization and also under several different sets of storage conditions. Therefore, the potato protein could be used as a protective carrier for hydrophobic nutraceuticals, suitable for enrichment of clear beverages and other food or drink products with beneficial impact on human health [143]. Lagaron et al. proposed a high degree of polymerization Agave fructans (HDPAF) as a novel encapsulating material for bioactive compounds [144]. It was demonstrated that they act as lyoprotectant agents on bovine plasma proteins and are able to cryoprotect food proteins. Fructans from *Agave tequilana* consist of a complex mixture of fructooligosaccharides. Electrospraying coating (solution and emulsion) was applied as the encapsulation technique. Encapsulated in the particles, they provide a protective effect of β-carotene by the nanocapsules (440 nm to 880 nm). Agave fructans from *Agave tequilana* Weber are considered microencapsulating materials of pitanga or Surinam cherry (*Eugenia uniflora* L.) juice by spray drying. Using FTIR, the interactions are detected by encapsulation of β-carotene with HDPAF. A comparison of nanocapsules of HDPAF and HDPAF/β-carotene obtained by the encapsulation process proved that the main differences between them are evident in the 1700–1800 cm^−1^ region because of the interaction of the C=O group of fructose molecules with β-carotene. The low intensity of β-carotene suggests that only a slight amount of this is located on the surface of the HDPAF nanocapsules because of a centripetal distribution of β-carotene, the highest concentration being in the core of the nanocapsule. This is probable explained due to the hydrophobicity of β-carotene, creating a barrier against oxygen and a protection against thermal decomposition processes, occurring mainly on the surface of the HDPAF nanocapsules [145]. β-carotene encapsulated in HDPAF by the EC method remained stable for up to 50 h of exposure to ultraviolet (UV) light.

Some examples of CRP with organic matrices are found in the review of Almasi and co. [10], the most recent being listed in Table 2.

The formation of inclusion complexes leads to significant changes in the solubility of the ACs, without any chemical modification to enhanced stability and bioavailability and also to reduced volatility of active compounds and their improved miscibility with hydrophilic matrices. Using β-CD inclusion complexes in the designing of CRP systems for the food packaging applications started in 2015, when Barba et al. [176] evaluated β-CD for the controlled-release of eugenol and carvacrol, observing their decreased release rate from the films. Other examples are given in Table 2.

### 3.5. Nanofillers and Nanocomposites

A nanofiller or nano-reinforcement should have at least one dimension less than 100 nm [177]. These ones are classified, as organic and inorganic, and according to their dimensions, such as [178,179,180]: (1) nanolayers, nanosheets, or nanoflakes (layered silicates such as nanoclay and nanolayered double hydroxides, zirconium phosphate nanolayer, starch nanocrystals); (2) nanotubes, having two nanoscale dimensions, such as nanotubes (carbon (CNT)), nanofibers, nanorods, and nanowhiskers (cellulose, chitin nanofibers, Ti nanotubes, halloysite nanotubes); and (3) nanoparticles (known also as 3D nanoparticles, isodimensional nanoparticles, nanogranules, nanocrystals or nanospheres with three dimensions in the nanometer scale (e.g., as silver NPs, and metal oxide NPs,TiO_2_, ZnO, Al_2_O_3_, SiO_2_, and chitosan NPs)). Incorporating nanofillers into packaging polymers and biopolymers improves the physical properties of the films, forming nanocomposites with unique properties. Two types of active nanocomposite have been produced: (1) nanocomposite film containing an active nanostructured material, such as antimicrobial NPs (Ag or TiO_2_ NPs); (2) a film having an active compound (such as antimicrobial or antioxidant chemicals) added for food preservation purposes or a nanostructured material (such as nanoclay, cellulose nanofiber, etc.), used as a reinforcing agent of the packaging material.

The nanomaterials can act as release-controlling devices in active nanocomposites, because of the strong interfacial interactions which occurred in the presence of NPs that can reduce the mobility of the polymer chains. The active molecules thus decrease the release of the active compound, or they crosslink the components and the tortuosity of the polymer network increases, preventing the burst effect. The nanomaterials with high surface area have the potential to form stronger interactions, especially nanolayers with the highest aspect ratio.

Some inorganic porous materials, such as various silica- or aluminosilicate-based materials/composites, clays, calcium carbonate, calcium phosphate, layered double hydroxides (LDHs), etc., have been also tested. For example, Ruiz-Rico et al. [181] investigated the controlled folic acid (FA) delivery and stability in fruit juices by its encapsulation into SiO_2_ mesoporous particles and obtained an improved vitamin stability and controlled release after consumption by modifying FA bioaccessibility.

The forms with lower toxicity, higher bioavailability, and controlled release, such as selenium NPs (SeNPs) and selenized polysaccharides, have been also studied [182].

The increased safety, shelf life, and stability of food products mainly for dietary supplements/nutraceuticals is necessary to be applied in practice. This is achieved when an appropriate stability of the active ingredient in the formulation is achieved at least until the date of consumption (expiration date). Nanoformulations of active compounds by incorporation in biodegradable nature- or semisynthetic-based nanocarriers, such as polymeric matrices, micelles, liposomes, nanoemulsions, solid lipid NPs, nanostructured lipid carriers, or appropriate inorganic matrices, are able to ensure not only an enhanced stability but also frequently controlled release of such nutrients. Supplementation of food products (e.g., bread, butter, yogurt, cakes, biscuits. etc.) or beverages (e.g., milk and juice) with individual healthy ingredients is the most convenient nanoformulation selected and used.

In other studies [183], electrically conductive thin layers with nanofiller contents of only 0.5 wt% were created using graphene nanoplatelets (GNPs) embedded in poly(ethylene-co-vinyl alcohol) (EVOH) nanofibers; these contact transparent EVOH films can be applied as smart tags in film interlayers. Wang et al. [184] demonstrated the potential of protein-inorganic hybrid nanoflowers to maintain or even increase the activity of the proteins and amplify the detection signals and colloidal semiconducting quantum dots (QDs) to detect bacterial cells or bacteria, cells, nucleic acids, molecules, ions, etc. [185]. Other examples are given in Table 2 and reference [10].

However, there are some risks in using nanomaterials at a large scale for preparing the packaging materials because of the dust inhalation [186]. The European Food Safety Authority (EFSA) has performed a risk assessment and has issued an opinion on each substance.

Factors which the authority take into account when making safety assessments include products’ toxicological properties and the extent to which they, or their breakdown products, could transfer into foods; migration of active or intelligent substances; and migration of their degradation and/or reaction products. In order for nanoformulations to enhance the bioavailability and increase the stability of individual active ingredients, all nanoscale materials applied in food industry should be used advisedly and only after in-depth investigation of cytotoxicity, due to possible increased nanosize-based toxicity effects (e.g., surface reactivity of NPs), because of unspecified toxic effects also in humans or animals. The influence of risk factors associated with their applications and possible adverse/hazardous effects to humans and animals can be studied according to the guidelines, regulations, and directives issued by the European Committee.

### 3.6. Modification of Polymer Matrix

Different modifications of the polymers have been proposed in order to achieve a more controlled diffusion of the AC from the packaging materials, concomitantly with improving their properties, the stability of active agents during the film forming process, and also leading to a more controlled release of the active compounds, such as chemical modification of polymers, irradiation, encapsulation, crosslinking, nanocomposite preparation, formation of inclusion complexes, lamination, etc.

Different examples on CRP systems obtained by either physical or chemical methods of modification are given in Table 3.

#### 3.6.1. Physical Methods/Technologies

Physical methods, such as corona discharge, ultraviolet (UV) radiation, gamma-ray at low doses, electron/ion beam, and plasma and laser treatments, can change the chemical structure of polymers-/biopolymers-based systems. These irradiations can lead to subsequent attachment of functional groups on a surface, allowing the material to immobilize enzymes or other bioactive species. Most are surface methods/technologies [209]. Plasma treatment is effective when it is applied during the processing of AP film, not on the previously prepared packaging material. Gamma irradiation and also electron beam irradiation are sometimes in nonsuitable conditions not only surface treatments, because the photon energy may be high enough to penetrate through materials.

Irradiation action on CRP has the ability to induce crosslinking in the polymer network being used for improving the physical properties of the films and also to tailor the release behavior of the active agents from CRP systems, because a more compact polymer network with a high/variable degree of crosslinking, causes the controlled release of the embedded active compounds.

The advantage of irradiation over the chemical processing of polymers is the occurrence of in situ crosslinking at lower temperatures in the solid-state of finished products, so that heating or melting of polymers is not needed and especially so that the use of hazardous chemicals is not required in this type of crosslinking [83,210]. Good adhesion and slower release of gallic acid was observed after plasma treatment. The prepared films exhibited strong activity against *E. coli* and *S. aureus* [189].

Heidemann et al. [211] applied atmospheric cold-plasma treatment to surfaces of poly (lactic acid) (PLA) and polycaprolactone (PCL) in order to improve the adhesion properties of a cassava starch film, thus obtaining multilayer films. These obtained multilayer films were characterized by significantly higher water-vapor barrier properties when compared to starch and by similar mechanical properties with PLA/PCL films. Using the same technique, Honarvar et al. [212] obtained an antimicrobial packaging of PP films coated with carboxymethyl cellulose (CMC) incorporated with Zataria multiflora essential oil. Another type of discharge, i.e., dielectric barrier discharge-generated atmospheric cold plasma, was used by another research group [213] for improving the adhesion and dyeing properties of chitosan-deposited ultrahigh molecular weight polyethylene (UHMWPE) fiber surfaces.

#### 3.6.2. Chemical Modification

Chemical modification consists in wet chemical techniques using chromic acid, potassium permanganate, or nitric acid (which are effective in surface modification by general oxidation reactions that generate carbonyl, hydroxyl, and carboxylic acid groups) [214], but sometimes, they are susceptible to overexposure, damaging the polymer surface, and leading to undesirable bulk property changes and crosslinking. By the intra- or inter-molecular crosslinking process, tridimensional networks are created by covalent or noncovalent bonds such as ionic interactions between divalent cations, hydrogen bonds, or hydrophobic interactions [194,215]. The ionic crosslinking of biopolymers occurs in mild reaction conditions, with a lower toxicity of reagents. Covalent bonds are created by irradiation or chemical reactions, using nontoxic crosslinking agents (bifunctional compounds such as aldehydes, formaldehyde, borax, epichlorohydrin, and citric acid). The crosslinking efficacy includes the following [216]: allows the controlled release of active compounds from package to food surface; reduces the mobility of the polymer chains; enhances mechanical and barrier properties of the systems; reduces water solubility and polymer swelling; increases the resistance to heat, light, chemicals and solvents; decreases the free volume, so increasing glass transition temperature; and retards the biodegradation of polysaccharide- and protein-based films and the disintegration of other polymers. The mechanisms proposed for controlled release are affected by decreasing the mobility of polymer chains, reducing segmental mobility, increasing the tortuosity by decreasing diffusion rate, decreasing the water sensitivity and swelling power, and decreasing the biodegradation and corrosion of polymer.

## 4. Processing Methods

The processing method strongly influences the release rates of CRP, which is why CRP films can be produced by traditional processing methods (cast film extrusion, blown film extrusion, and solvent casting, which is the use a smart blender attached to one or more extruders to produce polymer blend films with very different morphologies from those obtained by conventional extrusion methods) [217]. At the same time, special polymer blend morphologies of the LDPE/PP or LDPE/HDPE blends, from which ACs (as tocopherol) are released with desirable rates [218,219], were obtained by using a smart blending by manipulating polymer compositions and film morphologies.

Several techniques have been also used to encapsulate bioactive components for the food industry, such as extrusion methods [220], fluidized bed coating [221], spray cooling [222] or spray drying [223,224], and emulsion/nanoemulsion [6,225]. Because most EOs and NEs are volatile compounds, they require the use of manufacturing methods that are carried out at room temperature to preserve their original properties. In this sense, the electrospinning and electrospraying techniques are an emerging technology in the food packaging field.

Electrospinning, a modern technology, produces polymer nanofibers containing active compounds for the development of active food-packaging coating, encapsulation, and interlayer materials [226,227] with both antimicrobial and antioxidant activities [154,228,229]. The electrospinning technique facilitates the processing of the thermolabile substances. The basic set-up for electrospinning/electrospraying (Figure 2) consists of four main components: (1) a high-voltage source (1–30 kV), usually operated in direct current mode, though alternating current mode is also possible, (2) a blunt-ended stainless steel needle or capillary, (3) a syringe pump, and (4) a grounded collector in the form of a flat plate. The electrospraying process involves the application of a strong electrostatic field between two electrodes imposed on a polymer solution. When increasing the electrostatic field up to a critical value, charges on the surface of a pendant drop destabilize the shape of the solution from partially spherical to conical, i.e., the so-called Taylor’s cone effect. As the charged jet accelerates toward regions of lower potential, the solvent is evaporated [230]. Besides being a very simple technique, the solvent is evaporated at room temperature; thus, it constitutes an ideal method for protecting sensitive encapsulated ingredients.

To protect volatile compounds, both single and coaxial electrospinning is used.

## 5. Testing Methods for Performance of Active Food Packaging

There are many necessary testing methods to assess the performance of the active food packaging [116,118]. Some of them are shortly presented here.

### 5.1. General Methods of Characterization

These methods include optical properties (transparency and color), fiber diameter (μm), film thickness, morphological characterization by SEM, TEM, microscopies, structural information obtained by spectroscopies (UV, Infrared, NMR, etc.), gas chromatography, mechanical, rheological properties of the melts and solution properties prepared for electrospinning technique, thermal characterization by DSC, thermogravimetry TG/DTG, and chemiluminescence methods a.s.o. [6,232].

### 5.2. Surface Analysis

Surface properties. Water contact angle is an indicator of the surface hydrophilicity (θ < 65°) and hydrophobicity (θ > 65°) [233,234]. Surface tension and conductivity are general properties for surface characterization. Modern techniques giving rapid information without separation steps are, for food contact materials (polymers) and solid food simulant (Tenax^®^), direct thermal desorption techniques, such as atmospheric solids analysis probe (ASAP) MS, direct analysis in-real-time (DART) MS, desorption electrospray ionization (DESI) MS, liquid extraction surface analysis nano-electrospray, mass spectrometry (LESA-nESI-MS), micro Raman, and surface-enhanced Raman scattering (SERS) [235,236], X-ray photoelectron spectroscopy (XPS), Fourier-transform infrared spectroscopy (FTIR), atomic force microscopy, and scanning electron microscopy (SEM) [237].

### 5.3. Specific Methods

To assess bioactive compounds activity and packaging effects on food composition of bioactive compounds, one can use the determination of total phenolic content in vegetable oils by the Folin–Ciocâlteu reagent method and gas chromatography GC-MS [35].

*Barrier properties*. Barrier-properties performance for volatile compounds such as aromas, water vapors, and D- limonene are used as reference for WVT [109].

*Antioxidant activity* [238]. Oxygen-scavenging activity is determined by various tests, such as HPLC with UV (which was the most used technique), Trolox equivalent antioxidant capacity (TEAC) assay [239], *DPPH* [240], *ABTS^•+^* (2,2′-Azino-bis 3-ethylbenzthiazoline-6-sulfonic Acid) radical cation scavenging assay [35], by means of an OXY-4 mini (PreSens) multichannel fiber-optic oxygen meter [118], and others.

*Hexanal analysis.* Hexanal is the main product of linoleic acid oxidation. Therefore, it is an indicator of lipid oxidation, which has been used as a good marker to monitor many foods. It was done by headspace-solid phase microextraction (HS-SPME), coupled to a gas-chromatography–mass-spectrometry (GC–MS). The minimum inhibitory concentration (MIC) and minimum bactericide concentration (MBC) were determined.

### 5.4. Antimicrobial Activity

The antibacterial properties of SE and subsequently of the films were ascertained in triplicate against various microorganisms, such as the bacteria *Carnobacterium, Enterococcus, Lactobacillus, Lactococcus, Leuconostoc, Oenococcus, Pediococcus, Streptococcus, Tetragenococcus, Vagococcus,* and *Weissella*; each of them had a specific standard to be determined, such as *Staphylococcus aureus* (ATCC 6538P CECT240) and *Escherichia coli* (ATCC 25922), *Staphylococcus aureus, Listeria innocua*, and *Saccharomyces cerevisiae*; fungi (*Fusarium graminearum, Penicillium corylophilum, Aspergillus brasiliensis*, etc.), by the agar disc diffusion method modified by EUCAST (2010) [241], etc.

### 5.5. New Analytical Methods

Volatile compounds are analyzed by qualitative and quantitative analyses in samples of migration assays, which are performed using GC–MS (gas chromatography coupled to mass spectrometry), APGC–Q-TOF-MSE (atmospheric pressure gas chromatography coupled to quadrupole-time-of-flight mass spectrometry, elevated energy); and GC–Q-Orbitrap-MS (gas chromatography coupled to quadrupole-Orbitrap mass spectrometry), techniques which are able to detect the traces of analytes [242]. Solid-phase microextraction (SPME) sampling technique or stir-bar sorptive extraction (SBSE) are coupled to GC–MS system to increase sensitivity and also to analyze polymer pellets, flakes, or plastic itself [243,244]. In addition, the automatic multiple-dynamic hollow-fiber liquid-phase microextraction (HFLPME) and fabric-phase sorptive extraction (FPSE) coupled with GC–MS or liquid chromatography are very useful and sensitive techniques for determination of migrants from CRPs containing essential oils [245,246,247]. The combination of olfactometry with gas-chromatography–mass-spectrometry (GC–O–MS) allows analysis of the odorous compounds [248].

The interaction of plastics with components of the olive oils (tocopherols and polyphenols) submitted to accelerated conditions (up to 30 days, at 40 °C) was analyzed by 3D-front-face fluorescence spectra of olive oil [249].

*Non-Volatile Compounds.* For the profiling of non-volatile compounds at very low concentrations, liquid chromatography coupled to mass spectrometry (LC–MS) was used, with modern development of ultrahigh performance liquid chromatography coupled to quadrupole-time-of-flight mass spectrometry (UPLC–Q-TOF-MSE) [250,251]. Ion mobility separation (IMS), together with Q-TOF-MS, provides detailed structural insight for the analysis of nonvolatile compounds, and a more hybrid linear-ion-trap high-resolution mass spectrometry (LTQ-Orbitrap) combines a linear-ion-trap MS and the Orbitrap mass analyzer to obtain supplementary structural information about such compounds.

Table 4 summarizes some of the most-used analytical methods for detection of possible food contaminants.

Methods should be selected even for each contaminant or AC/analytical method, such as 2-propenoicacid,1,10-[2-[[3-2,2-bis[[(1-oxo-2-propen-1-yl)oxy]methyl] butoxy]-1-oxopropoxy]methyl]-2-ethyl-1,3-propanediyl]ester from varnished PP/vion IMS (Vion mobility quadruple) Q-TOF-MS; mono-2-ethyloxoexyl adipat from PLA/APGC–Q-TOF-MS (MS—ultrahigh performance liquid chromatography with electrospray ionization coupled to quadruple-time-of-flight with mass detector; LC–Q-Orbitrap-MS—liquid chromatography coupled to quadrupole-Orbitrap combined with mass spectrometry; cyclic oligomers (phthalic acid (PA) or/and diethylene glycol) from biodegradable polylactic acid (PLA), polyamide (PA), PET, PU were detected with migration tests, using ultrahigh performance liquid chromatography coupled to quadruple-time-of-flight with MS [253,254].

Antimicrobials, such as some essential oils (EOs) used in CRP, produce new substances as bioconversion products, such as methyl eugenol, styrene, and linalool oxide formed from cinnamon essential oil by *Aspergillus flavus* strain [255]

*Fatty Acids Oxidation.* These are extracted and then analyzed by capillary GC–MS. Thiobarbituric acid reactive substances to assess oxidation of lipids. Unsaturated fatty acids are oxidized to form odor-free, tasteless hydroperoxides. Then, they are decomposed to flavorful secondary oxidation products, which are mainly aldehydes, such as hexanal, 4-hydroxynonenal (HNE), and malondialdehyde (MDA) [256]. The most common method to determine MDA in foods is the spectrophotometric measurement of the colored adduct of MDA with 2-thiobarbituric acid (TBA).

### 5.6. Sensory Evaluation

Tasting is made by at least 10 experts and others. A quantitative descriptive analysis (QDA) should be performed to evaluate negative odorous attributes, as those from starch-based polymers. In this case, headspace solid-phase microextraction gas chromatography with olfactometry coupled to mass detector (HS-SPME-GC–O-MS), when the following analytes can be identified with specific odors: trimethylamine (fish), 1-octen-3-one (mushroom), sotolon (spices, licorice), nonenal (cucumber, fruit), eugenol (clove, honey), p-vinyguiacol (clove, curry), etc. [257].

The inorganic nanoparticles in solutions of food simulants are analyzed by inductively coupled with plasma mass spectrometry (ICP–MS). Two different modes can be used: single-particle mode (SP-ICP–MS) or ICP can be connected to the field-flow fractionation (FFF) technique that provides an additional separation step in respect with the size of nanoparticles. The shape, size, and image of nanoparticles can be observed using scanning electron microscopy (SEM), transmission electron microscopy (TEM), or atomic force microscopy (AFM), but these techniques have limitations at low concentrations of nanoparticles in case of migration assays when they have poor sensitivity [258].

### 5.7. Release Properties—Migration Methods

The migration levels in food are determined by one or more of the following methods: (1) accelerated migration studies conducted with food-simulating liquids under the most severe anticipated conditions of use, which is used to determine migrant concentrations in food; (2) the assumption of 100% migration to food, using actual use or residue levels; and (3) mathematical modeling of mass transfer from polymers to food, based on a thermal-processing-extended storage scenario, using actual use or residue levels.

Simulants are food substitutes which reflect the properties of the food stuffs which come into contact with the product or component. There are six different simulant types specified for use in overall migration testing, according to EU Regulation 10/2011 [259]: **simulant A**—10 percent ethanol (*v*/*v*), **simulant B**—3 percent acetic acid (*w*/*v*), **simulant C**—20 percent ethanol (*v*/*v*), **simulant D1**—50 percent ethanol, **simulant D2**—vegetable oil, **simulant E**—poly (2, 6-diphenyl-p-phenylene oxide), particle size 60–80 mesh, pore size 200 nm. Simulants A, B, and C are assigned for foods that have hydrophilic properties, covering aqueous, acidic, and alcoholic foods. Simulants D1 and D2 are for foods that have lipophilic properties and cover both dairy and nondairy fatty foods. Simulant E is assigned for testing specific migration into dry foods. Other selected simulants can be found in the study of Bailey et al. [260]. Once the appropriate simulant has been selected, the testing conditions (temperature and exposure period) need to be decided upon; there are seven specified overall migration limit testing conditions.

The release of catechin from the bilayer systems was studied in fatty-food simulant D1, and it was expressed as the amount of catechin and epicatechin released from the inner layer to the food simulant [107].

When applying the analytical procedures for migration tests, especially for antimicrobial-packaging analysis, new substances coming from biotransformation are present in the medium. Microorganisms from food can interact with essential oils used as the active agent in antimicrobial packaging and can produce new substances such as metabolites. Such results are important in the case of both real food and/or using food simulants for migration assays. The analysis of food becomes difficult due to the complexity of the matrix.

## 6. Mechanisms of Active-Compound Release

Between migration and sorption processes, an equilibrium is established during the lifetime of a commercial packaged foodstuff. Therefore, the limits of these processes must be measured for a specific food, an appropriate food simulant, and food packaging. The compatibility between the aroma and the food or the food simulant is the main contributing factor to the partition equilibrium describing the extent of food/packaging interactions. The measurement of liquid/vapor equilibrium can be used to compare the effectiveness of food simulants as substitutes of a particular food product and also for the selection of the appropriate simulant [261]. The migration of catechins and caffeine from active packaging with green tea for meat preservation is analyzed by UPLC–Q-TOF-MS when the main active agents were detected, but a further study in depth, searching the individual catechins, showed nine different compounds in 10% ethanol and 95% ethanol. All of them were coming from green tea, which is food, and are accepted as active packaging, according to the regulation 450/2009/EU [71,262].

Nanosilver and other metals are easily oxidized to ions but can re-form nanoparticles at slightly reductive conditions, e.g., at sample preparation and pretending particle migration. At cutting edges, the particles may be released due to weak binding to the surface. Nanoparticles which are completely encapsulated in the host polymer matrix do not have a potential to migrate into food. Thus, consumers would not be exposed to nanoparticles from food-contact polymers when those are completely embedded in polymer and the contact surface is not altered by mechanical surface stress during application. Migration modeling shows nanomaterials are immobilized in a polymer matrix [263]. Another problem consists in the detection of possible food contaminants from the packaging origin on the quality and safety of fresh food [252]. Food migrants/contaminants can be classified as intentionally added substances (IAS) to improve quality and safety of packaging and food, such as AC, antioxidants, antimicrobials (acetic acid, chitosan, catechin, gallic acid, lysozyme, and nisin), nanoparticles (see text and tables above), and non-intentionally added substances (NIAS), such as impurities from chemicals or primary materials; contaminants in the case of recycled materials; degradation products from polymer matrix and polymer additives; products of reaction between polymer components or reaction of polymer with food; monomers, oligomers, and printing inks, polymer degradation products, and aromatic volatile compounds (the last may compromise the safety and organoleptic properties of food). Removal of unwanted damaging substances from food is necessary, because they can have adverse effects on human health, due to migration of chemical substances from packaging origin (which are different types of polymers, paper, metal, and glass) to food structure. The transfer of migrants into food depends on packaging material type [264].

For directly incorporated food AP systems, three types of release mechanisms have been proposed [15,265]: (I) Diffusion-induced release of the AC through the microporous or macroporous structure of the synthetic and water-resistant polymer matrix towards the food; (II) Swelling-induced release. Because incorporated AC has a low diffusion coefficient when the system is placed in a fluid and the compatible polymeric matrix usually based in moisture-sensitive packaging materials (such as protein- or polysaccharide-based films) swell and the diffusion coefficient of the AC increases, diffusion takes place; (III) Disintegration-induced release: This takes place because of the degradation, cleavage, or deformation of polymer. This type of release occurs in biodegradable or reactive nonbiodegradable polymers, such as polyanhydrides, poly(lactide), poly(lactide-co-glycolide), etc.

### 6.1. Mathematical Models to Analyze Release Kinetic Data

The direct-type migration tests using analytical methods is time-consuming and, sometimes, not very efficient. That is why mathematical models have been used as predicting tools to understand migration characteristics and also providing insights for the design and performance evaluation of multilayer films [237].

The release of active compounds from a polymeric packaging film or food simulant involves three steps [266]:(1)molecular diffusion of bioactive compound (BC) towards the film/food interface,(2)mass transfer across the interface, and(3)dispersion into food or desorption into package headspace.

The first step is considered the slowest one; it determines the release rate, being described by two mathematical models based on diffusivity phenomena and partition coefficients [267,268,269]. The diffusivity shows the rate of migration of BC within the packaging film, while the partition coefficient shows the amount of BC released to medium, at equilibrium.

The most popular mathematical models to describe the release behavior are derived from one-dimensional Fickian diffusion, using differential equations applied in appropriate initial and boundary conditions or assumptions, such as the following: No structural change occurs in the polymer film during the release process; the active compound in the film is homogeneously distributed initially; initial concentration of the active compounds in food is zero; partition coefficient and diffusivity are constant at a given temperature; interactions between food simulant and film are absent or negligible; and no degradation of active compound occurs. Partition coefficients (K_f,p,_ K_p,f_) are defined as:(1)$$K_{f,p}=\frac{C_{f,\infty }}{C_{p,\infty }}\;\mathrm{or}\;K_{p,f}=\frac{C_{p,\infty }}{C_{f,\infty }}$$
where *C*_*f*,__∞_ and *C*_*p*,__∞_ are the concentrations of the active compound in food (*f*) and in package (*p*) at equilibrium, respectively.

The release curves of the active constituents of powdered-rosemary alcoholic extract by migration into 50% ethanol solution at 40 °C, as a food-simulant medium, from PLA/PEG/CS-based films, prepared by melt mixing, are represented in Figure 3. The release profiles of the active components of the rosemary from rosemary-powdered ethanol extract into 50% ethanol solution as a food simulant describing migration phenomenon into selected food simulant show an overall similar release behavior from the studied films, dependent on samples composition.

For example, the samples PLA/PEG/0.5R and PLA/PEG/3CS/0.5R show a fast release in the first 30 h; then, after 70 h the equilibrium is reached in the case of migration from plasticized PLA and from the biocomposite containing 3 wt% CS. At a content of 6 wt% CS, the migration is much slower and a more gradual release for entire studied period, and the equilibrium is reached after a longer period of about 350 h, in higher quantity until the end of 14 days. An amount of ~61% is released compared to 47% from the plasticized PLA/PEG/0.5R sample and 51% from PLA/PEG/3CS/0.5R sample. Therefore, the increase of CS content in the sample’s composition favors a gradual release without reaching a plateau even up to 10–12 days.

For short-term migration, defined as the time for which m_t_/m_∞_ < 0.6, a simplified migration model derived from Fick’s second law is applied, which considers diffusion as the main process governing the release of the active component, which occurs from both sides of the film, as described by Equation (2) [270]:(2)mtm∞=4D·t/l212
where *D* is the diffusion coefficient and *l* is the film thickness. A plot of m_t_/m_∞_ versus *t^1/2^* should yield a straight line, from which the diffusion coefficient can be obtained. The partition coefficient can be defined also as the ratio of the migrant concentration in the film (*C_f,_**_∞_*) to the migrant concentration in the food-simulant system (*C_s,_**_∞_*) at equilibrium (Equation (3)) [37,271]:(3)Kp=Cf,∞Cs,∞

The evaluated kinetic parameters are given in Table 5.

Sodium benzoate (SB) inhibits the growth of potentially harmful bacteria, mold, and other microbes in food, thus determining spoilage. It is particularly effective in acidic foods. It was recently tested in CRP based on multiple layers of chitosan, SB alginate beads in the inner layer, and alginate films [272]. The dependence of the release profiles on film thickness is represented in Figure 4. Kinetics of the first order, using the following model, was proposed to describe the two-stage release kinetics:(4)$$M_{t}/M_{\infty }=M_{1}\left(t\right)/M_{\infty }+M_{2}\left(t\right)/M_{\infty }$$
where *M_1_(t)* is the amount of antimicrobial released at the first stage of the release kinetics, and *M_2_(t)* is the amount of antimicrobial released in the second stage.

Lee and Yam [273] also developed a numerical [274] model to identify the target release kinetics of antioxidants and the optimum loading of antioxidant in packaging film, based on the reaction kinetics of lipid oxidation. The model was developed by combining a differential equation of hydroperoxides data. The release kinetics of the AC can be analyzed using the following equations:(5)$$\frac{M_{f,t}}{M_{f,\infty }}=1-\sum_{n=1}^{\infty }\frac{2\alpha \left(1+\alpha \right)}{1+\alpha +\alpha^{2}q_{n}^{2}}exp\left(-\frac{Dq_{n}^{2}t}{L_{p}^{2}}\right)\;(\mathrm{Model}\;1)$$
(6)$$\alpha =K_{fp}\frac{V_{f}}{V_{p}}$$
(7)$$tanq_{n}=-\alpha \cdot q_{n}$$

When the volume of food simulant is much larger than the volume of the film (i.e., *V_f_* >> *V_p_*), Model 1 can be simplified to the more commonly used Model 2, where *V_f_* is the volume of food simulant, *V_p_* is the volume of package (film), *K_fp_* is the partition coefficient, q_n_ values are the nonzero positive roots of $tanq_{n}=-\alpha \cdot q_{n}$, *M_f,t_* is the mass of the active compound migrated to the food simulant at time *t*, *M_f,_**_∞_* is the mass of the active compound in the food simulant at equilibrium, *Lp* is the film thickness, *D* is the diffusivity of the active compound in the film, and *t* is the time.
(8)$$\frac{M_{f,t}}{M_{f,\infty }}=1-\overset{\infty }{\underset{n=1}{\sum }}\frac{8}{{\left(2n-1\right)}^{2}\pi^{2}}exp\left(\frac{-D{\left(2n-1\right)}^{2}\pi^{2}t}{4L_{p}^{2}}\right)\left(\mathrm{Model}\;2\right)$$
with simplified form for short time when *M_f,t_/M_p,0_* ≤ 0.6,
(9)$$\frac{M_{f,t}}{M_{p,0}}=\frac{2}{L_{p}}{\left(\frac{Dt}{\pi }\right)}^{0.5}\;(\mathrm{Model}\;3)$$
where *M_p,0_* is the initial loading of the active compound in the package. *D* can be estimated from the slope of the plot of *M_f,t_/M_p,0_* versus *t^0.5^*. This model assumes total migration of the active compound to the food simulant (i.e., *M_p,0_ = M_f,1_*). Before applying these equations, it must verify if the model does fit the data, and the release kinetics follows Fickian diffusion, and/or some of the models’ mentioned assumptions are accomplished. If the data do not fit, a non-Fickian model accounting for the swelling effect must be used. These models provide useful insights of the release of the active compound to food simulants, but usefulness is quite limited for describing and predicting the release to real foods [15,269]. Software for migration tests was elaborated [275]. Some kinetic parameters obtained using these models are summarized in Table 6 [276].

### 6.2. Concept of Target Release Rate

The design of CRP systems is based on the concept of target release rate, defined as a release rate or a range of release rates of active compounds to effectively inhibit microbial growth or lipid oxidation at a target [285,289]. This depends on the food composition, the package, the distribution, and the storage conditions.

This definition was first presented considering the maximum inhibition of lipid oxidation in linoleic acid which provides different release rates, as in the case of the tocopherol, where the release rates cannot be too fast or too slow. A too-fast release caused more tocopherol released than linoleic acid free radical produced, and the excess tocopherol formed dimers or other products, making it not available for later stage of lipid oxidation; on the other hand, a too-slow release did not supply sufficient tocopherol for inhibiting lipid oxidation. This concept was further developed by Balasubramanian [289], who found that the initial amount of antimicrobial released in the inherent lag period of the target microbial needs to be greater than the minimal inhibition concentration (MIC), in order to achieve a high bio-efficacy, and a further inhibition. The new developed mathematical model—Model 4—takes the antimicrobial efficacy (MIC) and the microbial growth kinetics (lag period) into consideration, correlating them with the release kinetics of the antimicrobial from the film (diffusivity). Using this release rate model, one can identify the optimum release-rate profile based on the requirement of the food product that can be achieved using a smaller amount of antimicrobial.
(10)$$\frac{MIC_{IL}\times V_{f}}{M_{p,0}}=\frac{2A}{V_{p}}\sqrt{\frac{D\times t_{lag}}{\pi }}\;(\mathrm{Model}\;4)$$
where *MIC_IL_* is the minimum inhibitory concentration for an initial microbial load, *M_p,0_* is the initial amount of antimicrobial in the packaging film, *t_lag_* is the time taken for the organism to increase by 1 log, *V_f_* is the volume of food, *V_p_* is the volume of the package, *A* is the surface area of the polymer, and *D* is the diffusivity.

Target release profile is in form of a quantifiable parameter, such as diffusivity. It was found that the diffusivity between 7.5 × 10^−12^ m^2^/s and 2.60 × 10^−13^ m^2^/s was needed to provide complete inhibition of the microorganisms for 24 h, when 0.2 g (1 mg/mL) was added to the polymer. The release rate of antimicrobials from the package during the inherent lag period of the organism must be equal or more than their MIC, to produce an effective inhibition of the organism over the desired shelf life. The model takes into account the antimicrobial efficacy and the microbial growth kinetics (lag period), and correlates them with the release kinetics of the antimicrobial from polymer (diffusivity). It was established that the controlled release may use only 15% nisin to achieve the complete inhibition of *M. Luetus,* being smaller than instant addition of 100% nisin.

## 7. Different Indicators Used in Food Packaging

### 7.1. IOSP

IOSP systems contain small labels or tags, printed onto or incorporated into food packaging materials, in order to acquire information about the food’s quality, store that information, and transfer it to manufacturers, the retailers, or the consumers [17,290]. These indicators are classified in direct and indirect indicators [7], respectively. Direct indicators (i.e., maturation, freshness, pathogen indicators, etc.) monitor or sense storage conditions like pathogenic bacteria, volatile compounds, aroma compounds, biogenic amines, ATP degradation products, etc. These indicators offer information on microbial growth and spoilage, freshness, and edibility. Direct indicators must be usually placed inside the primary packaging, for having direct contact with the atmosphere surrounding the food or with the food itself.

On the other hand, indirect indicators (i.e., time–temperature indicators, gas indicators, radiofrequency identification tags) offer information on the storage conditions throughout the supply chain; these indicators indirectly evaluate the effects of the environment surrounding the food on the shelf life and quality of food, which might lead to a hidden danger for consumers, especially for children and the elderly [291]. 

### 7.2. TTIs

Because temperature is one of the main causes of food degradation during transportation, handling, distribution, storage, and consumption, temperature monitoring is very important to provide consumers with necessary information about food quality and safety throughout the process of food circulation. TTIs are generally attached onto individual consumer packages or shipping containers, and, depending on their capabilities, they can be classified into three categories [292], i.e.,(1)critical temperature indicators, which only show whether a product has been exposed to a temperature above, or sometimes below, a reference temperature;(2)critical TTIs, which indicate the cumulative effect of the time–temperature changes on product quality or safety and when a product has been exposed to a temperature above a reference temperature;(3)full history indicators, which provide a continuous monitoring of the manner in which the temperature varies with time throughout a product’s history.

TTIs allow the identification of irreversible responses (such as enzymatic or biological changes) induced when the food is exposed to a higher temperature. Between all TTI types, most of electronics-based TTIs have relatively high precision, are environmentally friendly, and can be recycled. This type of TTI can present a warning about the quality of a product by using a thermal sensor that converts temperature signals to electrical signals, after which the electrical signals are converted to a final visual output. Electronics-based TTIs are, however, generally expensive, due to the complexity of the involved read-out devices.

For other types of TTIs (such as nanoparticle-based, enzyme-based, chemistry-based, and biology-based), an irreversible color change is the main way to determine the thermal history of the product. These types of TTIs have a lower cost, are more convenient to read, and are smaller than electronics-based TTIs.

In the case of nanoparticle-based TTI, the size, shape, and surface morphology of metal nanoparticles change is based on the time/temperature history. Nanoparticles exhibit an irreversible color change when exposed to a particular temperature for a given time. Gelatin/AuNP (gold nanoparticle) [293] and alginate/AuNP [291] systems were used as indicators for thermal history.

In enzyme-based TTIs, the hydrolysis reaction of an enzyme with a substrate causes different degrees of color change, depending on the real time–temperature history. The observed color of a TTI can offer information on the cumulative effect of time and temperature, the information being used to implement a dynamic evaluation of the product’s remaining shelf life. 

Chemical TTIs are based on many different chemical reactions, such as polymerization, photochromic, and oxidation reactions, which induce different changes in the TTI color. Some examples of chemistry-based TTIs include Fresh-Check^®^, HEATmarker^®^ (NJ, USA), and OnVu^TM^ (Ciba Specialty Chemicals, Inc., Basel, Switzerland). 

### 7.3. RFID

RFID is based on wireless communication (magnetic field or electromagnetic wave) which can provide real-time information about temperature, relative humidity, and nutritional and supplier information while moving the product from the supplier to the consumer. Even if RFID tags are now considered to be a replacement for barcodes, their usage is still limited, due to their relatively high cost (approximately 0.2 to 0.3 USD per tag) [7].

RFID tags can be classified, according to the power supply mode, as:

**Passive**, which do not contain onboard power sources and are powered by electromagnetic induction in magnetic fields, which is produced near the reader. When compared with the other two types of RFID tags, the passive ones have a relatively short reading distance but a high operational life, making their small dimension and low cost be other advantages [294].

**Semipassive**, which have a local power source that is used only for powering the chip, being inactive most of the time. This fact increases the lifespan of this type of tags.

**Active**, which have an embedded battery that is used to power the chip and to broadcast signals to the reader. Compared with the other two types of tags, these tags have the widest reading range (more than 50 m), and many tags can be read simultaneously. Due to their high cost and to their lifespan depending on the battery life, the use of the active tags is limited.

### 7.4. Gas Indicators

The gas composition within food packaging changes due to several factors, such as the activity of the food product (i.e., respiration and transpiration of fresh horticultural products, as well as spoilage due to microorganisms), the nature of the package (i.e., the gas permeability), and the environmental conditions (temperature or package leaks). This gas composition significantly influences the integrity, shelf life, quality, and safety of packaged food products [290]. In order to obtain information on the changes in gas composition inside the package, different types of gas indicators are used in the form of package labels or package printing films that detect oxygen, ethanol, hydrogen sulfide (H_2_S), water vapor, carbon dioxide, or other gas components. In general, depending on the changes in gas composition, a gas indicator also provides an irreversible and visible color change on the packaging. 

As an example of gas indicator, one can mention the reversible oxygen indicator produced by DryPak Industries, which changes the color from pink to blue when the oxygen concentration exceeds 0.5%. Another oxygen indicator is developed by Mitsubishi Gas Chemical Company [7]. 

One of the new trends in improving the indicators used in food packaging is to integrate gas indicators into RFID tags. To obtain such a system, gas indicators have to be read automatically and changed into electrical signals.

### 7.5. Direct Indicators

After the food is packed up, volatile compounds can be generated within the package, due to enzymatic reactions, microbial growth, or chemical changes in the fresh food product. The quality and safety of packaged food products can be evaluated directly based on the levels of these volatile compounds. The principle on which direct indicators are based is to evaluate the quality and safety of the food product by finding the concentrations of the generated volatile compounds. Information obtained using direct indicators is more precise when compared with that obtained with the indirect ones.

Another type of a direct indicator is the pathogen indicators, used for detecting the presence of contaminating bacteria or pathogens, especially in meat products [290]. Even if very few commercial pathogen indicators are available for intelligent packaging in the global market, one can mention the Food Sentinel System™ (Pasadena, CA, USA), a system that utilizes barcode-based biosensors [295]. Biochemical reactions of a specific antibody with the target pathogens (such as *Salmonella* spp., *Escherichia coli*, and *Listeria monocytogenes*) are the basis of the operating principle of this system. The antibodies react with the target pathogen, causing the formation of a localized dark bar on the barcode, which becomes unreadable upon scanning. Another pathogen indicator is Toxin Guard^TM^ developed by Toxin Alert (Ontario, Canada). This one incorporates antibodies into polyvinyl chloride or polyolefin packaging films in order to detect pathogens [296].

## 8. Legislation for Using Packaging Materials

Legislation for using food packaging is periodically elaborated or updated, mainly at a national level but also at European (European Food Safety Authority) and worldwide levels. It refers both to main materials, additives, and bioactive compounds which can be tested to be food-contact safe and non-harmful to consumers, according to the safety standard and to environmental assessment (EA) for packed food and with no negative impact to the environment after applying the procedure for waste recovery/recycling. All food packaging materials must be rigorously tested by food safety agencies such as the U.S. Food and Drug Administration (FDA), the Brazil National Health Surveillance Agency (ANVISA), and the European Commission (EC), which are responsible for ensuring the safety of food packaging materials and additives before they can be used in food [297]. The US Food and Drugs Administration (FDA) has environmental responsibilities started under National Environmental Policy Act (NEPA) of 1969 [260].

Most food multilayer packages are either incinerated or landfilled [298], a fact that can be a serious problem for the environment and health. Hence, the application of biodegradable polymeric materials such as PLA, starch blends, and polyhydroxyalkanoates in food packaging has gained more attention in recent years [299]. In general, the reusability of composite films involves collecting, sorting, and recyclability. Since the preparation of multilayers involves the usage of adhesives for lamination of interlayers, recycling of these materials involves additional steps of delamination [300]. Apart from such adhesives, multilayer structure formation involves lamination with various other additives such as inks and colorants, increasing the level of difficulty of the recycling process. Physical or chemical delamination methods are used for recycling the multilayer packaging [237].

Toxicology testing and carcinogenic bioassays for food-contact substances are also necessary, based on dietary concentration (DC) and on corresponding estimated daily intake (EDI) values. Special attention should be given to the following categories: (1) substances used in the production and processing of food and which are not intended to remain with food, (2) processing aids used in the production of food-packaging material and which are not intended to remain as components of the finished packaging, and (3) components of food-packaging material present at greater than 5% by-weight (wt%) of the finished packaging.

In 1972, the European Commission (EC) drew up a broad program of action designed to harmonize all existing national laws in the field of materials intended to come into contact with food (especially plastics, and paper, ceramics, rubber, etc.) [274]. The 4th Amendment (Directive 2007/19/EC of March 30, 2007) specified rules with restricted phthalates, additives, functional barriers, and many others. Overall and specific migration limits, toxicological data, and mutagenicity studies are also required.

European Commission Regulation No 10/2011 [301] specified the plastic materials and articles intended to come into contact with food, the rules of migration experiments, and also a list of substances, together with their specific limits. The toxicological threshold is recommended by the European Food Safety Authority (EFSA).

## 9. New Trends and Necessary Developments

The purpose of all existing studies was to maintain the quality and safety and prolonging the shelf life of foods and beverages by slow releasing of the active compounds in a controlled manner. It is very important for CRP to know which is an optimum concentration profile of the active compound in dependence on time of the food-product distribution and storage conditions. This profile varies due to the food deterioration kinetics. Each profile is needed both for simulants and real foods. For this, are necessary experiments both for different active agents and foods and food simulants with a variety of profiles and systems with a slower and sustained release, which are not still studied. Until now, mainly liquid simulants in well-stirred systems have been studied. It is known that the release of the active compound to real foods is much slower than to liquid food simulants. If the active compound is released too fast, its concentration is not enough to prevent food deterioration. The results obtained in food simulants are not very useful to create kinetic models in order to predict the release of the active compounds into real-food products, because real-food products are much more complicated than food simulants. The new biobased packaging films should have competitive advantages over cellophane, (commercialized over a century ago), before the commercialization of synthetic polymers. Cellophane has superior mechanical, barrier, and optical properties, in comparison to most new biobased materials recently proposed. However, the price of cellophane in the last decades is too high in comparison with synthetic polymers, while its production is considered to be environmentally unfriendly as for production of other biobased polymer films, because they involve the use of toxic chemicals and/or unfriendly environmental solvents.

These studies might also lack the method of the film preparation, as a solvent casting method at laboratory scale, which is not similar to any industrial production technology. In addition, commercial equipment for producing biobased film needs to be developed. Development of a smart extruder to obtain controlled composition of the CRP and also of the electrospinning technology is necessary.

The most useful procedure to incorporate active compounds into the package usually facilitates unintended migration of other packaging additives into food, especially for the systems involving micro- and nano-encapsulation of active compounds, as some of the encapsulates may migrate with active compounds into the food. The possible migration of undesirable compounds may cause product safety and regulatory compliance issues and can affect the transfer of CRP technology from a bench to industrial scale.

It is a growing interest to develop controlled-release packaging for improving food safety and quality, as proven by the big number of publications in last decades. This is because the aim of this packaging technology is to deliver active compounds to the food in a controlled manner, providing a more effective and safe inhibition of food deterioration than using the traditional method of instant addition. This innovative technology passed the early development stages with significant results to improve food safety and quality. Its commercial success requires industry participation, regulatory approval, and market acceptance. 

## Figures and Tables

**Figure 1 molecules-26-01263-f001:**
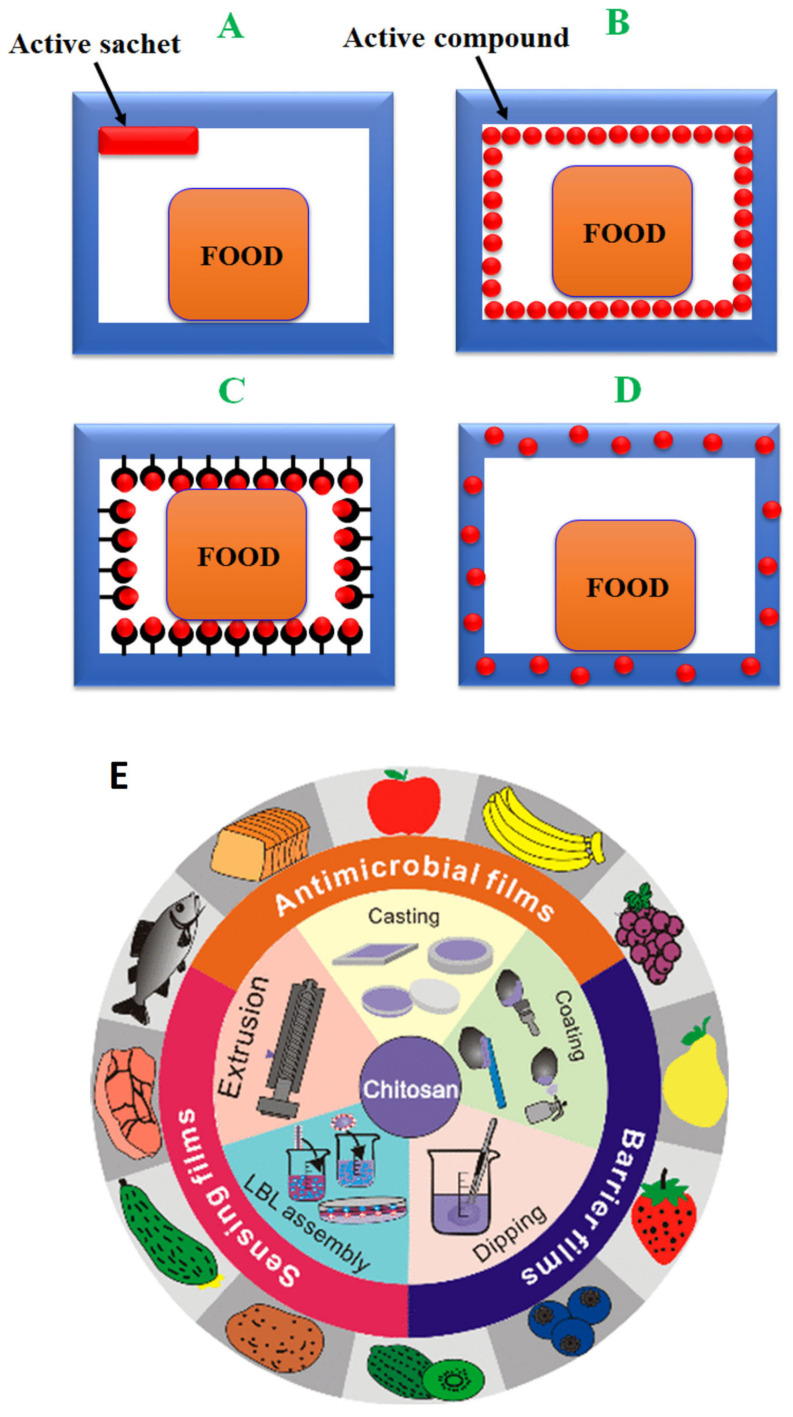
Types of active-packaging systems [8,10]. (**A**) using the active sachets (oxygen and moisture absorbers, and ethanol vapor generators), (**B**) coating an active compound (heat-sensitive active agents or those incompatible and immiscible with the polymer matrix) onto the polymer, (**C**) immobilizing the active compound on the polymer surface (the presence of functional interacting groups on both the active agent and the polymer is necessary). The strong bonding of active compounds onto polymers allows slow release into the food, and (**D**) direct incorporation into the polymer matrix, which ensures high resistance to processing conditions of the polymer, no adverse effect on the polymer properties, and slow release to food. Use of bioactive polymers, such as chitosan, exhibits inherently antimicrobial activity in composites or coating (**E**).

**Figure 2 molecules-26-01263-f002:**
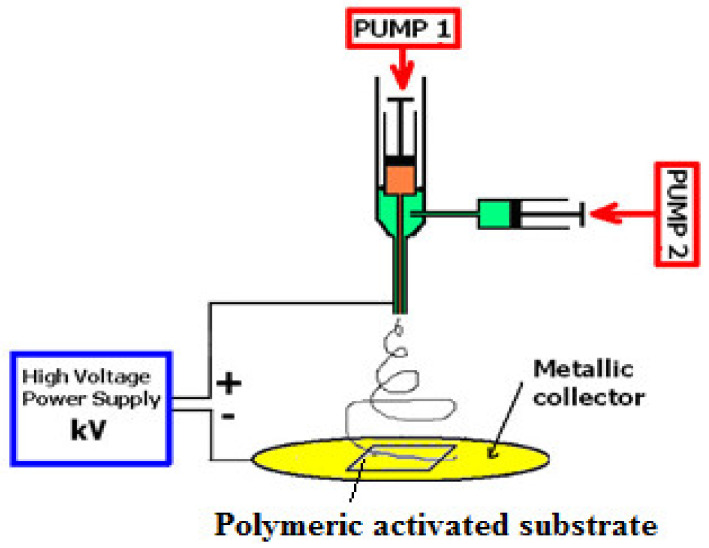
Coaxial electrospinning [209,231].

**Figure 3 molecules-26-01263-f003:**
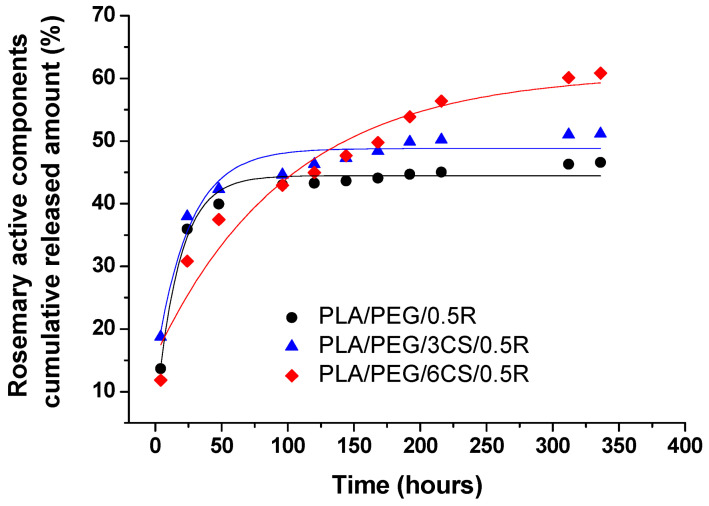
Release profiles of the active components from rosemary-powdered ethanol extract into 50% ethanol solution as a food simulant [37].

**Figure 4 molecules-26-01263-f004:**
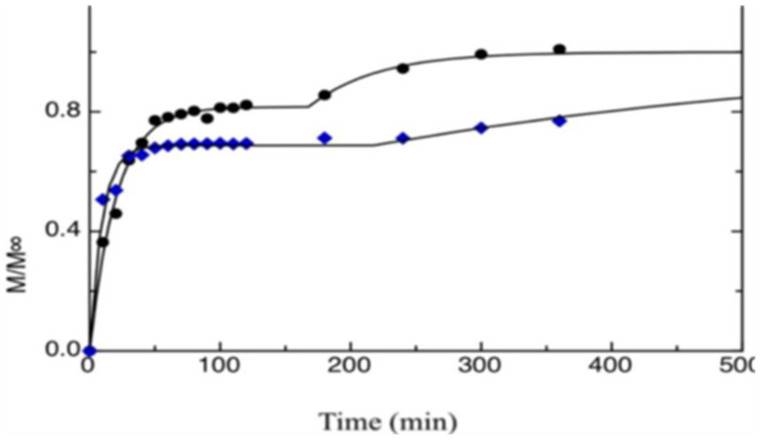
Release kinetic curves of sodium benzoate (SB) in water from both MULT-THIN (•) and MULT-THIK (◊) films, during the first stage of release [272].

**Table 1 molecules-26-01263-t001:** Examples of multilayer CRP.

Active Compounds (AC)	Packaging Structure	Effectiveness	Reference
Natamycin and/or nisin	Polyvinylidene chloride (PVDC) (active) or nitrocellulose (NC) coated on PE	Inhibition of selected microorganisms in cheese	[88]
Nisin	HPMC (active) coated on PE	Preservation of food shelf life	[89]
Nisin	Cellulose coated on PE	Inhibition of L. monocytogenes in tofu	[90]
Nisin or nisaplin	Ethylcellulose (EC)/HPMC (active layer)/EC	Food simulant analysis: EC was effective to delay release of nisin	[91]
Nisin or nisaplin	polyvinylchloride (PVC) (active layer) coated on low-density polyethylene LDPE	Inhibited microbial growth on cheese	[92]
Lysozyme	Layer by layer assembled chitosan organic rectorite and sodium alginate coated on cellulose acetate	Effective to inhibit growth of *Escherichia coli* and *Staphylococcus aureus*, and extend shelf life of pork for 3 days	[93]
Sodium benzoate and potassium sorbate	Pectin, pullulan, and chitosan as edible coating	Effective to reduce weight loss, fruit softening, delayed color degradation and titratable soluble solid in strawberry	[94]
Ascorbic acid and tocopherol	Tetraethyl orthosilicate and a mixture of alcoxysilane containing organic moieties (active) coated on polyamide-polyethylene (PA/PE)	Inhibition of oxidation shown by ferric reducing antioxidant power assay in EtOH	[95]
Sesamol or BHT	(1) LLDPE/HDPE (active)/HDPE, (2) HDPE /HDPE (active)/ethylene-vinyl acetate (EVA), (3) HDPE/HDPE (BHT)/EVA	Effective to inhibit lipid oxidation in linoleic acid and breakfast cereal	[66]
Cinnamon oil active with or without encapsulation in polyvinyl alcohol (PVA)	PVA coated on polypropylene (PP) and laminated with LDPE	Effective to repel instar larvae in a retail box containing flour	[96]
Chitosan oligomer	Thin layer of chitosan oligomers sandwiched in thermoplastic corn starch film	Effective to inhibit growth of yeast and mold in strawberries, ricotta, and flavored breads	[97]
Catechin (1–3%) antioxidant	Bilayer based on poly(3-hydroxybutyrate-co-3-hydroxyvalerate) (PHBV)/plasticized electrospun poly(lactic acid) (PLA)/ poly(3-hydroxybutyrate) (PHB) nanofiber blends. 15 wt% of oligomeric lactic acid was added as a plasticizer to increase stretchability	Bilayer films showed appropriate disintegration in compost conditions in around three months. Thus, they show their potential as biobased and biodegradable active packaging for fatty food products	[98]
Potassium sorbate	LDPE/polyamide (PA)/LDPE	The film samples within the liquid medium with low pH values had the highest diffusion coefficient	[99]
Scrophularia striata Boiss. extract (SE)	Bacterial cellulose (BCel)/BCel/BCel	Release rate and diffusion coefficient of SE in 95% ethanol simulant significantly decreased by lamination. Also, the dependency of SE release to temperature decreased after lamination	[100]
Quercetin	HDPE/LDPE/LDPE-EVA	The diffusion coefficient for films decreased by increasing the EVA amounts in inner layer, from 30% to 50%. The antioxidant activity values of the films were also enhanced as the EVA amount increased	[101]
Gallic acid (GA)	Polylactide (PLA)/PLA	The release rate of GA from the bilayer PLA films rapidly increased during the first 5 h of immersion. The PLA multilayers presented a high sustained release of GA, having the capacity to deliver the bioactive for over 1000 h	[102]
Tea polyphenols	Polypropylene (PP)/PVA/PP	Microporous PP films with different pore size were used as the inner layer. The diffusion coefficient for the films increased with the increase of pore size. TP release rate could be controlled by adjusting the pore size of the inner layer	[15]
Tea polyphenols	Zein/zein-gelatin/gelatin	Multilayer films exhibited prolonged release manner in comparison to mono- or bi-layer films	[103]

**Table 2 molecules-26-01263-t002:** CPR systems with organic matrices.

AC	Packaging, Structure/Preparation Method/Composition	Improvements	Reference
**Lipid, Polysaccharide, and Protein-Based Systems**			
Encapsulated oregano EO	Soluble soybean polysaccharide films/ Pickering emulsion stabilized by complex coacervates of acid-soluble soy protein and soluble soybean polysaccharide	By encapsulation, was immobilized and held more EO in the film matrix; films exhibited prolonged antimicrobial activity	[146]
Hesperidin Pickering emulsion stabilized with chitosan NPs	Activated gelatin film	Good compatibility and better antioxidant activity were the achievements of hesperidin encapsulation before loading to film	[147]
Oregano EO, resveratrol	Pectin/nanoemulsion	Nanoactive film showed the best performance on pork loin preservation, ascribed to the smaller particle size with enhanced preservative of EO	[148]
Nanoemulsion and Pickering emulsion-stabilized marjoram EO	Pectin-based active film	Whey protein isolate (WPI)-inulin microcapsule was more effective than nanoemulsion; release controlling of marjoram EO	[149]
Copaiba oil	Pectin/Nanoemulsion	Good compatibility of oil with film matrix was observed by nanoemulsion formation. Improved antimicrobial activity against *S. aureus* and *E. coli*	[150]
Encapsulated cumin seed EO	Pectin	Active papers containing WPI-inulin stabilized EO had the maximum effect on controlling the microbial growth and lipid oxidation of beef hamburger in comparison to free and nanoemulsion stabilized samples	[151]
**Cyclodextrin (CD) Inclusion Complexes**			
Tymol/β-CD complex	Thymol inclusion complex was incorporated into gelatin film’s casting solution	Release rate of thymol decreased after complexation and sustained release (235 h) of thymol was achieved by incorporating β-CD/thymol inclusion complexes into the gelatin films	[152]
-Gallic acid -allyl isothiocyanate -quercetin/β-CD complex	-PLA electrospun nanofibers/β-CD complex -PVA nanofibers / -polyacrylic acid (PAA) nanofibers/β-CD complex	Improvement in preservation rate, stability, and antibacterial activity	[153,154,155,156]
Tea tree oil (TTO)/β-CD complex	Poly(ethylene oxide) (PEO)	The release efficiency of the antibacterial agent from PEO nanofibers and the antibacterial activity of PEO nanofibers were improved. The highest antibacterial activity was observed against *Escherichia coli*	[157]
Curcumin/β-CD complex	Gelatin	Slower release of curcumin and better preservation of color and polyphenol contents of apple juice were observed by using curcumin/b-CD complex	[158]
Citral and transcinnamaldehyde/β-CD complex	EVOH	Low diffusion coefficient and high equilibrium concentration of active compounds was observed. The shelf life of beef was extended by inclusion complex formation	[159]
Carvacrol, oregano, and cinnamon EO/β-CD complex	Cardboard tray	Beneficial effect of active compounds on cherry tomato and bell pepper was maintained until day 24 and 18, respectively. Decay incidences reduced significantly	[160,161]
**Nanocomposites**			
Nisin	Starch/ Halloysite nanotubes 1D	Swelling of film decreased and its antimicrobial activity increased in the presence of halloysite	[162]
Clove EO	Soy protein isolate / MMT1D	MMT decreased the release of clove EO and prolonged its antimicrobial and antioxidant activity over time without observing the diffusion of the clay’s own metals	[163]
Carvacrol	Thermoplastic starch/neat and modified MMT1D	Organo-modified MMTs were more effective that Na-MMT in the release controlling of carvacrol, having a higher intermolecular affinity	[164]
*Origanum vulgare* ssp. gracile and Carum copticum EOs	Chitosan/cellulose and Lignocellulose nanofibers 2D	The active film had high antioxidant and antimicrobial activity, showing the release-controlling effect of CNF and LCNFa on the bioactive compounds from films	[165,166]
Cinnamon EO (CEO)	Sodium caseinate/cellulose nanofiber 2D	The CEO release ratio decreased by the addition of cellulose nanofiber (CNF). The increasing effect of temperature on CEO release decreased by incorporation of CNF	[167]
Cinnamon EO	Zein/Chitosan NPs 3D	The co-presence of CEO and CSNPs provided stronger inhibitory effects on *E. coli* and *S. aureus*, due to the enhanced delivery of EO, by loading in CSNPs	[168]
Thymol	CA films/AgNPs/gelatin-modified MMT	Swelling of the polymer structure, determining the diffusion of thymol to simulant media could be decreased in the presence of AgNPs and MMT. The release-controlling effect of MMT and AgNPs was different, and MMT had a higher delaying effect on the thymol volatizing process during film drying and storage, resulting in an upper remaining thymol amount in the films	[169]
Lythrum salicaria extract	Cellulosic nanomaterials	Immobilization of antimicrobial NPs or antimicrobial enzymes without a considerable decrease in their activity	[170,171]
Cumin EO	Sodium caseinate/TiO_2_ NPs 3D guar gum	The mechanical, barrier and antimicrobial properties of films increased synergistically when the TiO_2_ and cumin EO were added together	[172]
Rosemary EO; Betanin	Zein/j-carrageenan electrospun nanofibers incorporated with ZnO NPs; zein-sodium alginate nanofibers/TiO_2_ NPs	Mechanical and water barrier properties of nanofibers were improved and their hydrophilicity decreased by adding ZnO NPs. However, the presence of ZnO had no significant effect on the antioxidant and antimicrobial activity of the films.	[173,174,175]

**Table 3 molecules-26-01263-t003:** Examples of physical and chemical modification of polymer matrix to obtain CRPs.

Method Type	Polymer		AC		Modification		Effect on AC Release		Reference
**Physical methods**									
Plasma treatment	Polyvinyl alcohol (PVA) films		Nisin antimicrobial agent		Extent of nisin adhesion			A better adhesion of biologically active peptide-nisin to the polymer was obtained. It was also confirmed the nisin long-term stability on the PVA films	[187]
UV irradiation treatment	Ethylene -vinyl alcohol (EVOH) surface		Lysozyme		There were generated carboxylic acid groups, and the lysozyme was covalently attached to the functionalized polymer surface			Immobilized lysozyme reduces the growth of Gram-positive bacteria (*Micrococcus lysodeikticus* and *Listeria monocytogenes*), without migration of the lysozyme from the film; the enzymatic activity was decreased, being retained the entire antimicrobial activity.	[188]
Plasma treatment	LDPE		Gallic acid coated low-density polyethylene (LDPE) antimicrobial film		Surface functionalization			A slower release of gallic acid was observed after plasma treatment. The prepared films exhibited strong activity against *E. coli* and *S. aureus*	[189]
**Wet Chemical Methods**									
Chemically modified gliadin films	Protein			Cinnamaldehyde carrying lysozyme	As sustained release systems		The release rate of the antimicrobial agent was controlled by the reticulation of the protein matrix, and the degree of crosslinking led to the slower release of the AC		[190]
Chromic acid treatment and coating with clove EO	Linear low-density polyethylene (LLDPE) surface			Clove EO	Effective against selected pathogens, namely *Salmonella typhimurium* and *L. monocytogenes*.		The growth of pathogens was completely restricted in minced chicken samples on the fifth day of storage, and no further growth was detected during the 21 days storage period		[191]
**Crosslinking**									
**AC**		**System Polymer/Crosslinking Agent**				**Modification**			**Reference**
Lysozyme		Lysozyme enzyme from the gelatin films/Genipin less toxic than glutaraldehyde (GA)				The release kinetic profile of lysozyme from the neat gelatin films started with a burst effect, followed by subsequent slower release. After crosslinking, the burst release tended to be weakened and the cumulative release decreased			[192]
Maqui (Aristotelia chilensis) berry fruit extract or murta fruit extract, as a source of natural antioxidants		Methyl cellulose (MC)/glutaraldehyde				GA decreased water solubility, swelling and water vapor permeability (WVP) of the MC films, and the release of phenolic compounds decreased with the increase of the concentration of GA.			[193]
Lysozyme/Cinnamaldehyde antimicrobial and antifungal activity		Proteins–gliadin/crosslinking agent, due to its specific chemical structure consisting of a phenyl group attached; glandin				Gliadin film crosslinked with cinnamaldehyde preserved its integrity in water and prolonged the release of antimicrobial agent; a greater degree of crosslinking led to a slower release of lysozyme, i.e., the films with a loosely crosslinked structure released a greater amount of lysozyme, exhibiting a higher antimicrobial activity.			[190,194]
Natamycin		Ionic crosslinking between proteins (e.g., sodium caseinate) and polysaccharides (e.g., chitosan, alginate)				Water and mechanical resistance, barrier properties, cohesiveness, rigidity and also release controlling can be improved by the addition of Ca_2_ and barium ions. Interaction between natamycin and alginate chains in Ba-Ca films was stronger than that in the Ca-Ca and Ca-Ba films, therefore Ba-Ca films had the lowest natamycin’s diffusion coefficient.			[195,196]
Transglutaminase (TGase).Microbial, Vitamin B12		Enzyme-mediated crosslinking of the biodegradable films. Protein polymerization in presence of TGase to crosslink the casein system for the controlled release of VB12.				The casein hydrogel strength increased by the increase of TGase amount.			[197]
Doxycycline		Chitosan/gelatin-based hydrogels after TGase treatment				controlled release			[198]
Lysozyme		Sodium caseinate films activated with lysozyme with three crosslinking agents including glyoxal, calcium chloride, and TGase.				A slow release of lysozyme was achieved after the addition of glyoxal, by modulation in the antimicrobial activity against *M. lysodeikticus* and *Staphococcus aureus*. Crosslinking with glyoxal generated a caseinate network, able to release enzymatic activity in a gradual manner. However, calcium chloride and TGase caused stronger interactions in caseinate network, almost blocking enzyme release; adequate antimicrobial activity was not found in films			[199]
Nicotine		Chemical crosslinkers are polycarboxylic acids, able to crosslink carbohydratesby reaction with hydroxyl groups. Hydroxypropyl methylcellulose (HPMC) films were crosslinked with citric acid of nicotine.				Water insoluble films were created by crosslinking of HPMC with citric acid. At pH 2, pH 5.5 and pH 9, the released nicotine is diprotonated, monoprotonated and uncharged, respectively. The release rate tended to increase as the medium ionic strength increased			[200]
Ciprofloxacin		Recombinant silk-elastin like protein polymer/Ethanol or methanol vapor				Ethanol-treated film had the lowest swelling ratio, but the methanol-treated sample exhibited better release controlling of drug during 220 h			[201]
Rapamycin		Elastin-like protein polymers Rapamycin Genipin (solution), glutaraldehyde (solution and vapor) and disulfide (solution and vapor)				The best drug release-controlling effect was observed for disulfide, followed by glutaraldehyde in vapor state			[202]
Gentamicin sulfate		Sodium caseinate/ Alginate dialdehyde				Diffusion-controlled release of gentamicin was observed from the crosslinked films. The dynamic release was best interpreted by the Schott kinetic model			[203]
Tramadol		Gelatin/poly(ethylene glycol)/Citric acid				Slower release and maintaining of drug for a long time was achieved after crosslinking			[204]
Ketoconazole		β-cyclodextrin (β-CD)/ carboxymethylcellulose/Citric acid				β-CD and citric acid helped to minimize the burst effect and retarded the release of ketoconazole			[205]
Aloe vera gel extract		CMC-PVA/ Citric acid				Citric acid improved mechanical properties and diminished water solubility of the film. The shelf life of minced chicken meat was prolonged by using crosslinked active film			[206]
Esential Oil citral		Cashew gum-gelatin/ Ferulic acid				Antifungal activity of crosslinked film was observed for 6 days at the surface of bread, in comparison to 3 days of control			[207]
Jaboticaba anthocyanins		Casein hydrogel/Tgase				T-gase slower release rate of anthocyanins was observed for TGase treated hydrogels, at all studied pH values (2–7)			[208]

**Table 4 molecules-26-01263-t004:** Examples of analytical methods for substances released in different simulants [252].

Food Simulant	Method	Description Analytical	Comments
A	Ethanol 10% (*v*/*v*)	SPME-GC–MS; PLC–Q-TOF-MSE	Either by HS or total immersion modes
B	Acetic acid 3% (*v*/*v*)	SPME-GC–MS; UPLC–Q-TOF-MSE	Either by HS or total immersion modes
C	Ethanol 20% (*v*/*v*)	SPME-GC–MS; UPLC–Q-TOF-MSE	Either by HS or total immersion modes
D1	Ethanol 20% (*v*/*v*)	SPME-GC–MS; UPLC–Q-TOF-MSE	If SPME-GC–MS with total immersion of fiber is performed sample should be diluted at least five times.
D2	Any vegetable oil containing less than 1% unsaponifiable matter can be replaced by 95% ethanol and isooctane Liquid injection	GC–MS; UPLC–Q-TOF-MSE (reverse-phase column for 95% ethanol and normal-phase mode for isooctane)	If oil is used, it needs to be extracted. HS-SPME-GC–MS is also available for oil. When using 95% ethanol and isooctane, they can be concentrated under gentle stream of nitrogen, to gain sensitivity.
E	Poly(2,6-diphenyl-p-phenyleneoxide), known as Tenax^®^, particle size 60–80 mesh, pore size 200 nm	Liquid injection GC–MS; UPLC–Q-TOF-MS	Needs to be extracted with some organic solvent, like—for example—ethanol or methanol; they can be concentrated under gentle stream of nitrogen to gain sensitivity. Three sequential extractions are usually applied

SPME-GC–MS—solid phase microextraction gas chromatography coupled to mass spectrometry (MS) detector; HS—headspace; UPLC–Q-TOF-MS—ultrahigh performance liquid chromatography coupled to quadruple time-of-flight with MSE technology.

**Table 5 molecules-26-01263-t005:** Kinetic parameters of the bioactive compounds release from plasticized polylactic acid (PLA)-based materials containing chitosan (CS) and rosemary extract [37].

Samples	Peppas/Power Law Model				First Order Kinetic Model		Diffusion Model		K_p_
	n	R^2^	k × 10^3^ (h^−n^)	R^2^	k_1_ × 10^3^ (h^−n^)	R^2^	D × 10^−13^ (m^2^/s)	R^2^	
PLA/PEG/0.5R	0.37	0.99	85.12	0.98	5.20	0.84	1.70	0.98	1.06
PLA/PEG/3CS/0.5R	0.23	0.98	147.78	0.99	5.17	0.81	2.05	0.97	0.95
PLA/PEG/6CS/0.5R	0.38	0.99	72.92	0.99	3.70	0.92	1.05	0.99	0.64

**Table 6 molecules-26-01263-t006:** Some kinetics data obtained using the previous described mathematical models [276].

Active Compounds	Packaging	D × 10^−10^ (cm^2^/s)	K^n^	T (°C)	Release media	Reference
**Model 1**						
Potassium sorbate	LMP/CMC 8/2	0.0026	0.00062	4	95% EtOH	[277]
	LMP/CMC 4:6	0.0150	0.00505	25		
	LMP/CMC 4:6	0.0311	0.01758	60		
Eugenol	SPI	0.003	1155	5	Olive oil	[39]
Cinnamaldehyde	SPI	0.001	1445	5	Olive oil	
Eugenol	SPI	0.003	1155	5	Olive oil	
Isoeugenol	SPI	0.01	535	5	Olive oil	
Astaxanthin	LDPE	0.354	55.54		95% EtOH	[278]
Lysozyme	CA with various CA/H_2_O ratios and porosities	1.50–23.3	234–826	4	Water	[74]
L-ascorbic acid	CA with various CA/acetone/H_2_O ratios and porosities	1.67–15	169–2439	4	Water	[75]
Tocopherol	Ziegler-Natta LLDPE	4.2	83%	30	Coconut oil	[279]
**Model 2**						
Carvacrol	Soy-protein-coated paper, under various RH (60%- 80%)	0.0011–0.075	NA	5	H_2_O & n-pentane mixture	[280]
		0.0085–0.0878	(50/50)	20		
Quercetin	EVA	133	0.01	RT	95% EtOH	[39]
Tocopherol	HDPE/EVOH/LDPE (active)	0.234	27.91	20	Whole milk powder	[80]
Tocopherol	LDPE wt% various loadings of tocopherol	0.13–0.14	NA	5	Corn oil	[281]
		0.71–0.96		20		
		3.03–5.11		30		
Tocopherol	LDPE	0.264	NA		95% EtOH	[282]
Nisin	EVA	1130	NA	10	66% H_2_O, 32% PO w/2% emulsifier	[22]
Tocopherol	Ecoflex	983	NA	30	95% EtOH	[283]
BHT	PLA	19.04	12.61	43	95% EtOH	[284]
**Model 3**						
Clove essential oil H_2_O	EVA 0.25	0.25	NA	23	0.30	[61]
Tocopherol	LDPE	4.60	NA	40	95% EtOH	[285]
Cinnamaldehyde	Gliadin w/ various loading of cinnamaldehyde	0.0488–1.31	NA	20	Released into headspace	[286]
Quercetin	LDPE	1.15 × 10^−5^	001	RT	95% EtOH	[39]
Nisin	Acrylic polymer (active) coated paper	420	1.3% 10^5^	10	Water	[287]
Catechin	PLA	0.019	NA	40	50% EtOH	[288]

K is calculated by K_P,F._

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
