# Peer review of "Progresses in Food Packaging, Food Quality, and Safety—Controlled-Release Antioxidant and/or Antimicrobial Packaging"

_molecules, 2021, doi:10.3390/molecules26051263_

Round 1

Reviewer 1 Report

The article "Progresses In Food Packaging, Food Quality And Safety" presented for review is an extensive review of modern food packaging. The authors have included many aspects in it, ranging from the types of active packaging, through their composition and structure, types of active additives and indicators, controlled release of active substances, processing and testing methods, to the applicable legal acts.

 Each of the topics contains the necessary explanations and examples, which gives the reader a good understanding of this matter. The article is written in an interesting and clear way.

However, I would like to suggest the following changes or adjustments:

  1. I suggest you add a table of contents, it will simplify finding content in the article.
  2. Please correct the repeated numbering of paragraphs 6 and 7.
  3. Some typos need to be removed.
  4. Figure 1 E should be enlarged because it is difficult to read
  5. Line 545 "The low intensity of ß-carotene suggests" and 547 "Nanocapsules with a low surface intensity observed by FTIR" these sentences need to be improved as the intensity relates to characteristic bands of compounds, or their functional groups, but to the compounds themselves.
  6. Figure 3 is briefly mentioned in the text, but it is worth adding a comment or conclusion to it, e.g. which of these materials is better and why, or which release dynamics is more favorable / more useful from a practical point of view.

Author Response

ANSWERS TO REVIEWER 1

Manuscript ID: molecules-1096810

Type of manuscript: Review

Title: Progresses In Food Packaging, Food Quality And Safety – Controlled Release Antioxidant and/or
Antimicrobial Packaging

Reviewer #1:

General comments: Authors: Cornelia Vasile *, Mihaela Baican. The article "Progresses In Food Packaging, Food Quality And Safety" presented for review is an extensive review of modern food packaging. The authors have included many aspects in it, ranging from the types of active packaging, through their composition and structure, types of active additives and indicators, controlled release of active substances, processing and testing methods, to the applicable legal acts. Each of the topics contains the necessary explanations and examples, which gives the reader a good understanding of this matter. The article is written in an interesting and clear way. However, I would like to suggest the following changes or adjustments:

Comment

Answer

I suggest you add a table of contents, it will simplify finding content in the article.

Added

Please correct the repeated numbering of paragraphs 6 and 7.

Corrected

Some typos need to be removed.

Corrected

Figure 1 E should be enlarged because it is difficult to read

enlarged

Line 545 "The low intensity of ß-carotene suggests" and 547 "Nanocapsules with a low surface intensity observed by FTIR" these sentences need to be improved as the intensity relates to characteristic bands of compounds, or their functional groups, but to the compounds themselves.

This part (line 545 and 547) was re-written:  A comparison of nanocapsules of HDPAF and HDPAF/β-carotene obtained by the encapsulation process proved that HDPAF is the dominating component. The main differences between them are evident in the 1700–1800 cm−1 region because of the interaction of the C=O group of fructose molecules with β-carotene. The low intensity of β-carotene suggests that only a slight amount of this is located on the surface of the HDPAF nanocapsules because of a centripetal distribution of β-carotene, the highest concentration being in the core of the nanocapsule. This is probable explained due to the hydrophobicity of β-carotene,  creating a barrier against oxygen and a protection against thermal decomposition processes, occurring mainly on the surface of the HDPAF nanocapsules

Figure 3 is briefly mentioned in the text, but it is worth adding a comment or conclusion to it, e.g. which of these materials is better and why, or which release dynamics is more favorable / more useful from a practical point of view.

Information was completed (pp. 34 in the manuscript): The release profiles of the active components of the rosemary from rosemary powdered ethanol extract into 50% ethanol solution as a food simulant describing migration phenomenon into selected food simulant show an overall similar release behavior from the studied films, dependent on samples composition.  For example, the samples PLA/PEG/0.5R and PLA/PEG/3CS/0.5R show a fast release in the first 30 h, then after 70 h the equilibrium is reached in the case of migration from plasticized PLA and from the biocomposite containing 3 wt% CS. At a content of 6 wt% CS, the migration is much slower and a more gradual release for entire studied period and the equilibrium will be reached after a longer period of about 350 h, in higher quantity until the end of 14 days; an amount of ~ 61% is released compared with 47% from the plasticized PLA/PEG/0.5R sample and 51% from PLA/PEG/3CS/0.5R sample. Therefore, the increase of CS content in the samples composition favors a gradual release without reaching a plateau even up to 10–12 days.

 All the changes we have made to the manuscript are in red, and the paragraphs we moved from one section to another are in yellow. The references were renumbered after all the changes we have made.

Reviewer 2 Report

This manuscript pretends to review the last offer on smart packaging but it partially reviews active packages based on the controlled release of antioxidant and antimicrobial agents. Thus, in my opinion, this work requires a major revision to be publishable. In my opinion, the title should be changed to describe what the work actually reviews. A possibility could be: Food active packaging based on agent controlled release.

In the introduction, there are many deficient definitions. Smart packaging is a term that includes active packaging and intelligent packaging. The difference between active packaging and intelligent packaging is well defined in the European regulation. Intelligent packaging provides information on the product or on the product quality and safety. Active packaging systems are those that include substances deliberately added to the packaging which action improves the stability of food quality and safety. As the introduction reads, these packages are based on the release or on the retention of substances whose presence or absence is related to food quality and safety, or based on the direct contact with the food providing a condition that improves food stability. Thus, controlled release packaging are a part of active packaging.

The introduction also contains information which should be deleted such as paragraphs devoted to migration (line 39 to 72).

In section 3.1. the manuscripts reads “Common active compounds used in CRP include mainly antimicrobials for food safety and antioxidants  for food quality, oxygen or ethanol scavengers, and CO2 emitters.” Oxygen and ethanol scavengers, by definition, are not CRP systems. I agree with the authors that antioxidant and antimicrobial packaging are the two most relevant CRP systems, nevertheless, the treatment provided to these two types is too different. There are hundreds of articles focused on antimicrobial packaging  and the papers describes them with a paragraph and 5 references. On the contrary, papers focused on antioxidant packaging are less abundant but the reviews covers them exhaustively (31 references).

In paragraph 5.4. devoted to antimicrobial packaging, authors include hexanal analysis (lines 771 and ss). This paragraph should be moved to the previous packages since it is a test for antioxidant analysis.

In paragraph 5.5, authors describe diverse methods which are relevant in active packaging but also include methods to analyze migration which should be deleted (lines 778 to 791). Also diverse comments and data in tables of this section related to migration should be deleted.

Section 5.7 should be changed removing all aspects related to migration and the information related to active packaging can be postponed to section 6.

Section on intelligent packaging is too short and based on commercial products instead of scientific papers. According to my first comment, the review could be complete and relevant if the authors decide to focused on antioxidant and antimicrobial packaging based on controlled release.

Author Response

Manuscript ID: molecules-1096810

Type of manuscript: Review

Title: Progresses In Food Packaging, Food Quality And Safety– Controlled Release Antioxidant and/or
Antimicrobial Packaging

Reviewer #2:

General comments: This manuscript pretends to review the last offer on smart packaging but it partially reviews active packages based on the controlled release of antioxidant and antimicrobial agents. Thus, in my opinion, this work requires a major revision to be publishable

Comment

Answer

In my opinion, the title should be changed to describe what the work actually reviews. A possibility could be: Food active packaging based on agent controlled release.

New proposed title:  Progresses In Food Packaging, Food Quality And Safety.  Controlled Release Antioxidant and/or
Antimicrobial
Packaging

In the introduction, there are many deficient definitions. Smart packaging is a term that includes active packaging and intelligent packaging. The difference between active packaging and intelligent packaging is well defined in the European regulation As the introduction reads, these packages are based on the release or on the retention of substances whose presence or absence is related to food quality and safety, or based on the direct contact with the food providing a condition that improves food stability. Thus, controlled release packaging are a part of active packaging.

Definitions have been corrected (page 2 in the manuscript). Smart packaging is a term that includes active and intelligent materials and articles may be placed on the European market if they comply with the restrictions set out in Regulation (EC) 1935/2004, articles, 3, 4 and 15, and the European Regulation (EC) 450/2009. Intelligent packaging provides information on the product or on the product quality and safety. Active packaging systems are those that include substances deliberately added to the packaging which action improves the stability of food quality and safety, or based  on the direct contact with the food, providing a condition that improves food stability. Thus, controlled release packaging are a part of active packaging.

The introduction also contains information which should be deleted such as paragraphs devoted to migration (line 39 to 72).

These information was moved to CRP (page 5 in the manuscript)

In section 3.1. the manuscripts reads “Common active compounds used in CRP include mainly antimicrobials for food safety and antioxidants  for food quality, oxygen or ethanol scavengers, and CO2 emitters.” Oxygen and ethanol scavengers, by definition, are not CRP systems. I agree with the authors that antioxidant and antimicrobial packaging are the two most relevant CRP systems, nevertheless, the treatment provided to these two types is too different. There are hundreds of articles focused on antimicrobial packaging  and the papers describes them with a paragraph and 5 references. On the contrary, papers focused on antioxidant packaging are less abundant but the reviews covers them exhaustively (31 references).

Thanks for comment. More information has been included in text and also references, mainly reviews (pp. 6-7 in the manuscript).

An antimicrobial is an agent that kills microorganisms or stops their growth. Antimicrobial packaging is multifunctional by reducing harmful microbial activity in food, helps to increase food safety, reduces food wastage and improves food shelf life. Bio-based antimicrobial agents in packaging provide extra safety for health [Sung S.-Y.; Sin L.T.; Tee T.-T.; Bee, S.-T.; Rahmat, A.R.; Rahman, W.A.W.; Tan, A.-C.; Vikhraman M. Review. Antimicrobial agents for food packaging applications. Trends in Food Science & Technology 2013, 33, 2, 110-123.] Those for food preservation act to prevent growth of spoilage and pathogenic microorganisms. Antimicrobial agents are incorporated into polymer film/packaging to suppress the activities of targeted microorganisms, as against Listeria monocytogenes, Mycobacterium smegmatis (MTCC 943), Pseudomonas aeroginosa (MTCC 4676), Escherichia coli O157, Salmonella, Staphylococcus aureus, Bacillus cereus, Campylobacter, Clostridium perfringens, Aspergillus niger, Saccharomyces cerevisiae etc. Between the most used antimicrobials, one can mention: main natural compounds, as essential oils derived from plants (e.g., basil, thyme, oregano, cinnamon, clove, and rosemary), enzymes obtained from animal sources (e.g., lysozyme, lactoferrin), bacteriocins from microbial sources (nisin, natamycin), organic acids (e.g., sorbic, propionic, citric acid), naturally occurring polymers (chitosan and its derivatives) [Sahariah, P.; Másson, M. Antimicrobial Chitosan and Chitosan Derivatives: A Review of the Structure–Activity Relationship. Biomacromolecules 2017, 18, 11, 3846–3868], sodium benzoate etc., all approved to be used in contact with food [Buonocore, G.G. Del Nobile, M.A.; Panizza, A.; Corbo, M.R.; Nicolais, L.A general approach to describe the antimicrobial agent release from highly swellable films intended for food packaging applications. J. Controlled Release 2003, 90, 97-107].

Interest in “antimicrobial activity” of chitosan is huge evidenced in 2014 by over 1140 articles, with 740 of these published after 2010. Derivatization of chitosan is realized by acylation, carboxylation, alkylation, and quaternization in order to improve the water solubility, pH sensitivity, and the targeting in the antibacterial, sustained slowly release, targeting, and delivery system fields. Chitosan-derivatives present excellent antimicrobial activity due to permanent positive charge on nitrogen atoms side-bonded to the polymer backbone [Wang, W.; Meng, Q. ; Li, Q.; Liu, J.; Zhou, M.; Jin Z.; Zhao, K.Chitosan Derivatives and Their Application in Biomedicine. Review Int. J. Mol. Sci. 2020,21(2),487; https://doi.org/10.3390/ijms21020487; Martins, A.F.; Facchi,S.P.;  Follmann, H.D.M.; Pereira, A.G.B.; Rubira, A.F.; Muniz, E.C. Review, Antimicrobial Activity of Chitosan Derivatives Containing N-Quaternized Moieties in Its Backbone: A Review Int. J. Mol. Sci. 2014,15, 20800-20832; doi:10.3390/ijms151120800; Goya, R.C.; Morais, S.T.B.; Assis, O.B.G. Evaluation of the antimicrobial activity of chitosan and its quaternized derivative on E. coli and S. aureus growth. Revista Brasileira de Farmacognosia 2016,26,1 Curitiba Jan./Feb https://doi.org/10.1016/j.bjp.2015.09.010]

Other interesting polysaccharides used for antimicrobial packaging are alginates and carrageenan [Cha, D.S.; Choi, J.H.; Chinnana, M.S.; Park, H. J. Antimicrobial Films Based on Na-alginate and κ-carrageenan. LWT - Food Science and Technology, 2002, 35, 8, 715-719]. Nanoformulations of silver nanoparticles with cellulose, chitosan, and alginic acid biopolymers for antibacterial applications have been prepared [Alavi, M.; Rai, M, Recent progress in nanoformulations of silver nanoparticles with cellulose, chitosan, and alginic acid biopolymers for antibacterial applications. Applied Microbiology and Biotechnology 2019, 103, 8669–8676]. Incorporation of silver NPs as nanocomposite (NC) forms improved antibacterial activities of these polysaccharides. Antimicrobial packaging with lactic acid bacteria incorporated in alginate film matrix was found to control the growth of food-borne pathogens in ready-to-eat food [Rinaudo, M. Biomaterials based on a natural polysaccharide: alginate. TIP 2014, 17, 1, 92-96. https://doi.org/10.1016/S1405-888X(14)70322-5]

Antimicrobial agents from organic sources (as plant phenolics, carvacrol, thymol, bacteriocins) and monoterpene hydrocarbons (p-cymene and γ-terpinene) compounds which are present in oregano essential oils, citric acid (in green tea), (clove, oregano, and thyme), enzymes (lysozyme, lactoperoxidase, glucose oxidase,), polymers (chitosan and derivatives), organic acid (acetic acid, lactic acid, and benzoic acid), bacteriocins (nisin), and metal ions (zinc oxide zinc, silver nanoparticles, copper, palladium, and titanium  more stable at higher temperatures) were incorporated into the protein-based film [Said N. S. ; Sarbon N. M. Protein-Based Active Film as Antimicrobial Food Packaging: A Review. Chapter Active Antimicrobial Food Packaging in Books. Active Antimicrobial Food Packaging. 18 pag. Published: January 30th 2019 INTECHOPEN https://www.intechopen.com/books/active-antimicrobial-food-packaging/protein-based-active-film-as-antimicrobial-food-packaging-a-review]. Nowadays, protein-based film technology has emerged as one of the most extensively studied in food packaging sector, as it exhibits good mechanical, optical, and oxygen barrier properties. In addition, protein-based film also showed good compatibility to polar surfaces, while having effective control on the release of additives and bioactive compounds in food packaging system. Antimicrobial food packaging gained great interest due to high inhibition of microbial activity that helps in prolonging the shelf life of packaged food and enhancing the food’s safety, while improving the functionality of the films, and also they are biodegradable. However, biocomposite protein-based packaging films obtained by incorporation of antimicrobial agents might also require chemical, toxicological, and further test in securing more safe and approved products according to the standard food safety regulations while being able to deliver good means in protecting the safety and quality of packaged food.

In paragraph 5.4. devoted to antimicrobial packaging, authors include hexanal analysis (lines 771 and ss). This paragraph should be moved to the previous packages since it is a test for antioxidant analysis.

Moved (page 29 in the manuscript, section 5.3)

In paragraph 5.5, authors describe diverse methods which are relevant in active packaging but also include methods to analyze migration which should be deleted (lines 778 to 791). Also diverse comments and data in tables of this section related to migration should be deleted.

Lines 778-791 have been deleted.

Some comments have been deleted

Section 5.7 should be changed removing all aspects related to migration and the information related to active packaging can be postponed to section 6.

Lines 885-905 from section 5.7 have been moved to section 6 (pp. 32-33 in the manuscript)

Section on intelligent packaging is too short and based on commercial products instead of scientific papers. According to my first comment, the review could be complete and relevant if the authors decide to focused on antioxidant and antimicrobial packaging based on controlled release.

Title was changed

All the changes we have made to the manuscript are in red, and the paragraphs we moved from one section to another are in yellow. The references were renumbered after all the changes we have made.

Round 2

Reviewer 2 Report

The review has been improved. It can be published in this revised version